# Cloud-based framework for inter-comparing submesoscale permitting realistic ocean models

Takaya Uchida[1], Julien Le Sommer[1], Charles Stern[2], Ryan P. Abernathey[2], Chris Holdgraf[3], Aurélie Albert[1], Laurent Brodeau[4,5], Eric P. Chassignet[6], Xiaobiao Xu[6], Jonathan Gula[7,8], Guillaume Roullet[7], Nikolay Koldunov[9], Sergey Danilov[9], Qiang Wang[9], Dimitris Menemenlis[10], Clément Bricaud[11], Brian K. Arbic[12], Jay F. Shriver[13], Fangli Qiao[14], Bin Xiao[14], Arne Biastoch[15,16], René Schubert[7,15], Baylor Fox-Kemper[17], William K. Dewar[1,18], and Alan Wallcraft[6]

[1]Université Grenoble Alpes, CNRS, IRD, Grenoble-INP, Institut des Géosciences de l'Environnement, France
[2]Lamont-Doherty Earth Observatory, Columbia University in the City of New York, USA
[3]2i2c.org, USA
[4]Ocean Next, Grenoble, France
[5]Datlas, Grenoble, France
[6]Center for Ocean-Atmospheric Prediction Studies, Florida State University, USA
[7]Univ. Brest, CNRS, Ifremer, IRD, Laboratoire d'Océanographie Physique et Spatiale (LOPS), IUEM, 29280, Plouzané, France
[8]Institut Universitaire de France (IUF), Paris, France
[9]Alfred Wegener Institute (AWI), Helmholtz Center for Polar and Marine Research, Germany
[10]Jet Propulsion Laboratory, National Aeronautics and Space Administration (NASA), USA
[11]Mercator Ocean International, France
[12]Department of Earth and Environmental Sciences, University of Michigan, USA
[13]Oceanography Division, US Naval Research Laboratory, USA
[14]First Institute of Oceanography, and Key Laboratory of Marine Science and Numerical Modeling, Ministry of Natural Resources, Qingdao, China
[15]GEOMAR Helmholtz-Zentrum für Ozeanforschung Kiel, Germany
[16]Kiel University, Kiel, Germany
[17]Department of Earth, Environmental, and Planetary Sciences, Brown University, USA
[18]Department of Earth, Ocean and Atmospheric Science, Florida State University, USA

**Correspondence:** Takaya Uchida (takaya.uchida@univ-grenoble-alpes.fr)

**Abstract.** With the increase in computational power, ocean models with kilometer-scale resolution have emerged over the last decade. These models have been used for quantifying the energetic exchanges between spatial scales, informing the design of eddy parametrizations and preparing observing networks. The increase in resolution, however, has drastically increased the size of model outputs, making it difficult to transfer and analyze the data. It remains, nonetheless, of primary importance to

5   assess more systematically the realism of these models. Here, we showcase a cloud-based analysis framework proposed by the Pangeo Project that aims to tackle such distribution and analysis challenges. We analyze the output of eight submesoscale-permitting simulations, all on the cloud, for a crossover region of the upcoming Surface Water and Ocean Topography (SWOT) altimeter mission near the Gulf Stream separation. The cloud-based analysis framework: i) minimizes the cost of duplicating and storing ghost copies of data, and ii) allows for seamless sharing of analysis results amongst collaborators. We describe

10  the framework and provide example analyses (*e.g.*, sea-surface height variability, submesoscale vertical buoyancy fluxes, and

comparison to predictions from the mixed-layer instability parametrization). Basin-to-global scale, submesoscale-permitting models are still at their early stage of development; their cost and carbon footprints are also rather large. It would, therefore, benefit the community to document the different model configurations for future best practices. We also argue that an emphasis on data analysis strategies would be crucial for improving the models themselves.

## 1 Introduction

Traditionally collaboration amongst multiple ocean modelling institutions and/or the reproducing of scientific results from numerical simulations required the duplication, individual sharing and downloading of data, upon which each of the interested parties (or an independent group) would analyze the data on their local workstation or cluster. We will refer to this as the 'download' framework (Stern et al., 2022). As realistic ocean simulations with kilometric horizontal resolution have emerged (*e.g.*, Rocha et al., 2016; Schubert et al., 2019; Brodeau et al., 2020; Gula et al., 2021; Ajayi et al., 2021), such a framework has become cumbersome with tera- and peta-bytes of data needed to be transferred and stored as ghost copies. Nevertheless, a real demand exists for collaboration to inter-compare models to examine their fidelity and quantify robust features of submeso- and meso-scale turbulence (the former on the horizontal spatial scales of $O(10\,\mathrm{km})$ and latter on $O(100\,\mathrm{km})$; here on referred to jointly as (sub)mesoscale; Hallberg, 2013; McWilliams, 2016; Lévy et al., 2018; Uchida et al., 2019; Dong et al., 2020). The Ocean Model Intercomparison Project (OMIP), for example, has been successful in diagnosing systematic biases in non-eddying and mesoscale-permitting ocean models used for global climate simulations (Griffies et al., 2016; Chassignet et al., 2020).

Here, we would like to achieve the same goal as OMIP but by inter-comparing submesoscale-permitting ocean models, which have been argued to be sensitive to grid-scale processes and numerical schemes as we increasingly push the model resolution closer to the scales of non-hydrostatic dynamics and isotropic three-dimensional (3D) turbulence (Hamlington et al., 2014; Soufflet et al., 2016; Ducousso et al., 2017; Barham et al., 2018; Bodner and Fox-Kemper, 2020). Considering the enormous computational cost and carbon emission of these submesoscale-permitting models, it would also benefit the ocean and climate modeling community to compile the practices implemented by each modeling group for future runs. In doing so, we analyze eight realistic, submesoscale-permitting ocean simulations, which cover at least the North Atlantic basin, run with the code bases of the Nucleus for European Modelling of the Ocean (NEMO; Madec et al., 2019, https://www.nemo-ocean.eu/), Coastal and Regional Ocean COmmunity model (CROCO; Shchepetkin and McWilliams, 2005, https://www.croco-ocean.org/), Massachusetts Institute of Technology general circulation model (MITgcm; Marshall et al., 1997, https://mitgcm.readthedocs.io/en/latest/), HYbrid Coordinate Ocean Model (HYCOM; Bleck, 2002; Chassignet et al., 2009, https://www.hycom.org/), Finite volumE Sea ice-Ocean Model (FESOM; Danilov et al., 2017, https://fesom2.readthedocs.io/en/latest/index.html), and First Institute of Oceanography Coupled Ocean Model (FIO-COM, http://fiocom.fio.org.cn/). Considering the amount of data, however, the download framework becomes very inefficient. Therefore, we have implemented the 'data-proximate computing' framework proposed by the Pangeo project where we have stored the model outputs on the cloud and brought the computational resources adjacent to the data on the cloud (Abernathey et al., 2021a; Stern et al., 2022).

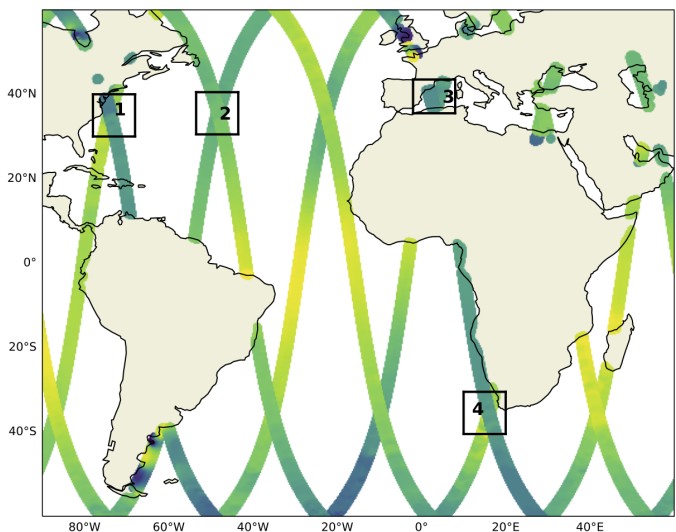

**Figure 1.** SWOT tracks during its calibration phase and strategic Xover regions in the Atlantic sector. The regions cover the Gulf Stream separation and its extension (Regions 1 and 2), western Mediterranean Sea (Region 3) and Agulhas Rings (Region 4).

Many of these simulations were developed ahead of the Surface Water and Ocean Topography (SWOT) satellite launch
(Morrow et al., 2019), now projected to be in November 2022, in order to allow for the instrumental calibration of SWOT
(Gomez-Navarro et al., 2018; Metref et al., 2020), and to disentangle the internal wave signals from (sub)mesoscale flows;
SWOT is expected to observe the superposed field of the two dynamics (Savage et al., 2017a; Torres et al., 2018; Yu et al.,
2021). During its calibration phase, SWOT will pass over the same site every day for six months and have tracks that will
cross over with each other. In order to showcase the data-proximate computing framework and its potential for collaborative,
open-source and reproducible science, we provide example diagnostics for one of the SWOT Crossover (Xover) regions around
the Gulf Stream separation (Region 1 in Figure 1). We leave the detailed diagnostics of (sub)mesoscale flows including other
SWOT-Xover regions and the potential impact of modeling numerics on the resolved flow for a subsequent paper.

The paper is organized as follows: We describe the data-proximate computing framework in section 2 and showcase some
example analyses using this framework in section 3. Cautionary remarks regarding sustainability into the future for open-source
reproducible science are given in section 4 and we conclude in section 5.

## 2   Data-proximate computing framework

In order for the data-proximate computing framework to work for collaborative, open-source and reproducible science, it
requires two components to work together simultaneously: i) public access to analysis-ready data, and ii) open-source compu-
tational resources adjacent to the data.

## 2.1 Analysis-ready cloud-optimized data

In the field of Earth Science, model outputs are often archived and distributed in binary, HDF5 or NetCDF formats. While we have greatly benefited from these formats, they are not optimized for cloud storage nor for parallelized cloud computing. However, as Earth Scientists, commonly, we do not possess the training in cloud infrastructure nor data engineering required to efficiently convert large scale archival datasets into formats which allow us to leverage the full performance potential of the commercial cloud. Data engineers, on the other hand, do not know the scientific needs of the data. In collaboration with Pangeo Forge (Stern et al., 2022, https://pangeo-forge.readthedocs.io/en/latest/), we have therefore, attempted to fill this niche by streamlining the process of data preparation and submission. To transform their data into analysis-ready cloud optimized (ARCO) formats, data providers (ocean modeling institutions in our case) need only specify the source file location (*e.g.*, as paths on an Ftp, Http or OPeNDAP server) along with output dataset parameters (*e.g.*, particular ARCO format, chunking) in a Python module known as a *recipe*. The recipe module, which is typically a few dozen lines of Python code, relies on a data model defined in the open source `pangeo-forge-recipes` package. Once complete, the recipe is submitted via a Pull Request on Github to the Pangeo Forge `staged-recipes` repository (https://github.com/pangeo-forge/staged-recipes). From here, Pangeo Forge automates the process of converting the data into ARCO format and storing the resulting dataset on the cloud, using its own elastically-scaled cloud compute cluster. The term "analysis-ready" here is used broadly to refer to any dataset that has been preprocessed to facilitate the analysis which will be performed on it (Stern et al., 2022). An example of such recipe for eNATL60 described in section 3 is given in Appendix A. We refer the interested reader to Abernathey et al. (2021a) and Stern et al. (2022) for further details on the technical implementation.

The crowdsourcing approach of Pangeo Forge, to which any data provider can contribute, not only benefits the immediate scientific needs of a single research project, but also the entire scientific community in the form of shared, publicly accessible ARCO datasets which remain available for all to access. This saves each scientist the cost of duplicating and storing ghost copies of the data and allows for reproducible science. The model outputs used for this study are stored on the Open Storage Network (OSN), a cloud storage service provided by the National Science Foundation (NSF) in the U.S. The surface data were saved hourly and interior data in the upper 1000 m as daily averages (due to cloud storage constraints). To facilitate the access of data from OSN, we have further made them readable via `intake`, a data access and cataloging system which unifies the API to read and load the data (https://intake.readthedocs.io/en/latest/overview.html). Namely, the API to read and load the data is the same for all of the data used in this project, regardless of its distribution format (*e.g.*, binary, HDF5 or NetCDF), because each of the datasets has been converted by Pangeo Forge into the cloud-optimized Zarr format (https://zarr.readthedocs.io/en/stable/), and subsequently catalogued with `intake`, prior to analysis. This is particularly beneficial for our case where we would like to systematically analyze multiple data collections. The entire process of zarrifying the data, fluxing them to OSN and cataloging scaled well for the four regions shown in Figure 1; the net amount of data stored on OSN as of writing sums up to $O(1\,\text{Tb})$. While the amount of $O(1\,\text{Tb})$ may seem small, the ARCO framework negates the generation and storage of ghost copies, and scales as the data size increases. Jupyter notebooks for the results shown in section 3, including the Yaml file to access data via `intake`, are given in the Pangeo Data `swot_adac_ogcms` Github repository (https://github.com/roxyboy/swot_adac_

ogcms/tree/notebook; a DOI will be added upon acceptance of the manuscript). Regarding LLC4320, the data were accessed via the Estimating the Circulation and Climate of the Ocean (ECCO) data portal. While there was no particular sub-setting applied to their dataset prior to analyses, the data portal and cloud-based JupyterHub being within geographical proximity (within the U.S.) facilitated the data access. The combination of `llcreader` of the `xmitgcm` Python package to access their data in binary format (as opposed to NetCDF) also enhanced the efficiency (Abernathey, 2019; Abernathey et al., 2021b).

## 2.2  Cloud-based JupyterHub

For data-proximate computational resources, we have implemented a JupyterHub, an open-source platform that provides remote access to interactive sessions in the cloud for many users (Fangohr et al., 2019; Beg et al., 2021), on the Google Cloud Platform (GCP). This infrastructure is run in collaboration with 2i2c.org, a non-profit organization based in the U.S. that manages cloud infrastructure for open source scientific workflows. Authentication for each user/collaborator on the JupyterHub is provided via a white-list of Github usernames, meaning that the hub can be accessed from anywhere and is not tied directly to an institutional account. This has allowed for real-time sharing of Python scripts amongst collaborators and exchanging of feedback on the analytical results we present in section 3. Cloud computing also offers the scaling of resources for improved I/O throughput and optimization of network bandwidth and Central Processing Units (CPUs).

## 3  Example analyses

The model outputs used for this showcase are from the eNATL60 (Brodeau et al., 2020), GIGATL (Gula et al., 2021), HYCOM50 (Chassignet and Xu, 2017, 2021), FESOM-GS, LLC4320 (Rocha et al., 2016; Stewart et al., 2018), ORCA36 (https://github.com/immerse-project/ORCA36-demonstrator), FIO-COM32 (Xiao et al., 2022), and HYCOM25 (Savage et al., 2017a, b; Arbic et al., 2018) simulations. The detailed configuration of each simulation is given in Appendix B. In order to motivate the reader on the necessity of inter-comparing realistic submesoscale-permitting simulations, we show in Figure 2 the surface relative vorticity normalized by the local Coriolis parameter on February 1, 00:00 from each model. Despite their similar spatial resolutions, the spatial scales represented vary widely across models. Submesoscale-permitting ocean modeling is in its early stage of development, and each modeling institution is still exploring best practices. Therefore, we did not specify an experimental protocol, as in OMIP, for the model outputs from each institution. Each model uses different atmospheric products and tidal constituents to force the ocean, and the initial conditions and duration of spin up all vary (Appendix B). Nevertheless, we should expect statistical similarity in the oceanic flow at the spatial scales of $O(10\,\mathrm{km})$ if the numerics are robust.

## 3.1  Surface diagnostics of the temporal mean and variability

In light of the SWOT mission, the primary variable of interest is the ocean dynamic sea level. The AVISO estimate of this quantity is called the Absolute Dynamic Topography (ADT), while the closely related model diagnostic is the Sea Surface Height (SSH) after correcting for the inverse barometer effect if atmospheric pressure variability was simulated. Technically,

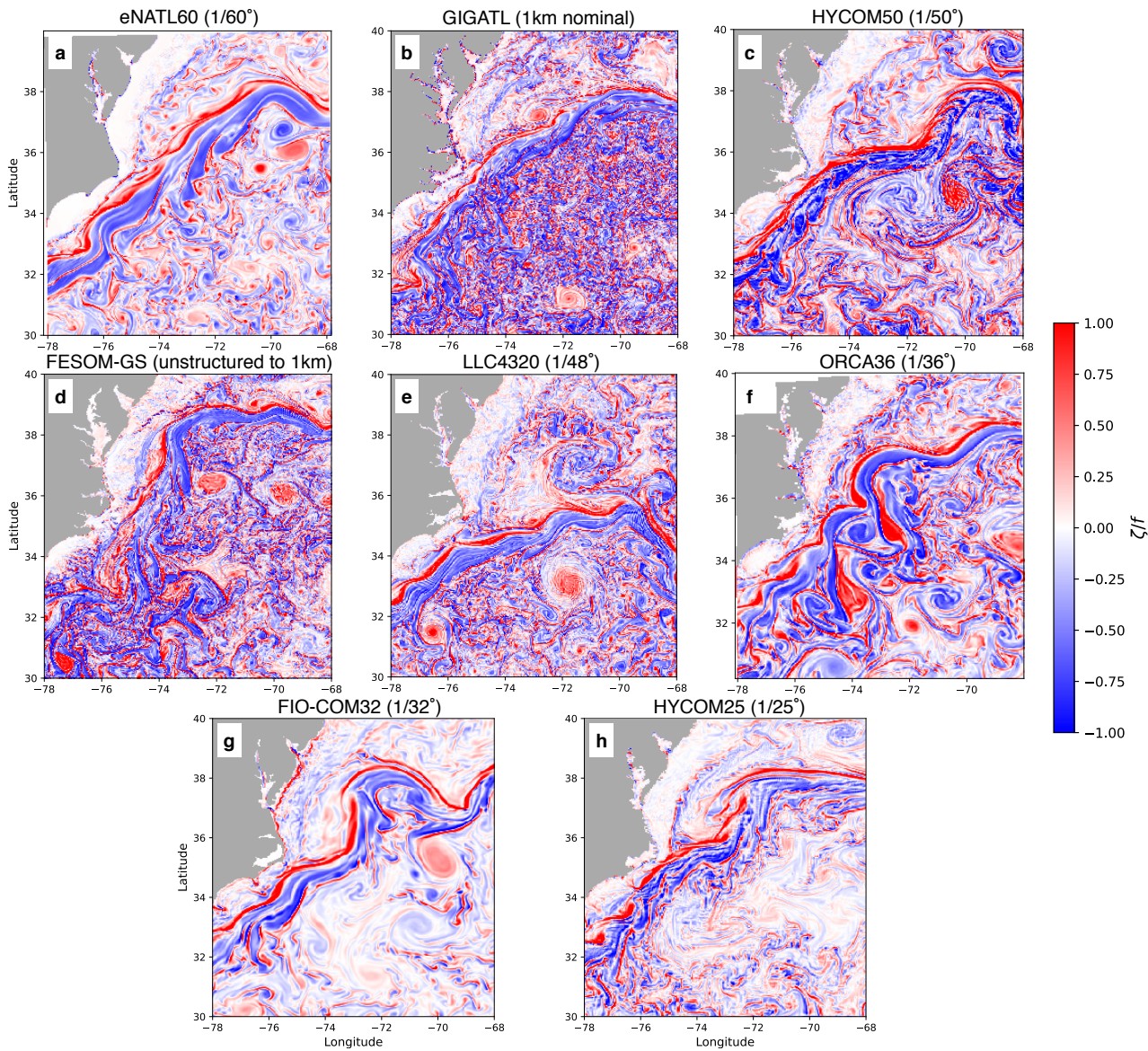

**Figure 2.** A snapshot of surface relative vorticity normalized by the local Coriolis parameter on February 1, 00:00 from each model in Region 1.

SSH is defined as the geodetic height of the sea surface above the reference ellipsoid, while ocean dynamic sea level (or ADT) is defined relative to the geoid, but in models typically the geoid and reference ellipsoid coincide so these two definitions are in practice the same (Gregory et al., 2019). Furthermore, in the specific comparisons made here, a regional average of the ocean dynamic sea level estimates is removed first, so that large-scale, slow changes (*e.g.*, ice sheet mass loss contributions) are excluded from the comparison. From an ocean modelling perspective, one of the key features to argue in favor of increasing

resolution in the North Atlantic has been the improvement in representing the Gulf Stream (GS) separation and path of the North Atlantic Current (Chassignet and Xu, 2017; Chassignet et al., 2020; Chassignet and Xu, 2021). In assessing the models, it is common to examine the mean state, which we define as the time mean, and variability about the mean. From the perspective of computational cost, the time mean of surface fields is the lightest as the reduction in dimension allows for the download framework where the collaborators can share the averaged data. Variability about the time mean requires access to the temporal

dimension, making the computational and data storage cost intermediate. We will further show in section 3.2 an example of 3D diagnostics of the submesoscale flow, which significantly increases the computational cost and burden of transferring data; the 3D diagnostics will highlight the strength of the data-proximate framework where we can consistently apply the same diagnostic methods across different datasets.

    In Figure 3, we show the time mean and temporal standard deviation of ocean dynamic sea level from the eight models in the

GS separation region. We also show the time-mean ocean dynamic sea level estimated as ADT from the Archiving, Validation and Interpretation of Satellite Oceanographic (AVISO) data for reference. We do not show the standard deviation for AVISO as the spatiotemporal interpolation and smoothing limit its effective resolution to $O(100\,\mathrm{km})$ and $O(10\,\mathrm{day})$ (Chelton et al., 2011; Arbic et al., 2013; Chassignet and Xu, 2017). We provide the modeled standard deviations of ocean dynamic sea level filtered in a manner similar to the smoothing that goes into the AVISO products in Appendix C. The GS in most models tends to separate

off of Cape Hatteras on the east coast of the U.S. consistent with AVISO (Figure 3a,c,g,i,k,o,t). In terms of the magnitude of mean SSH, HYCOM50 may be overestimating it relative to AVISO across the path of the separated Gulf Stream. The GS in LLC4320 tends to separate relatively southwards while in FESOM-GS separates northwards relative to AVISO observations respectively (Figure 3e,i). The separation in FESOM-GS may be closer to the observed state in 2014 (Figure 3s) rather than 2012, the actual year of model output. Regarding the standard deviation, while expected, it is interesting that the simulations

without tides (FESOM-GS and ORCA36; Figure 3f,l) show significantly lower temporal variability compared to the other models. The low variability in FESOM-GS could also stem from the lack of atmospheric pressure variation in their atmospheric forcing (Table B6). Although HYCOM25 is tidally forced, its standard deviation is relatively low (Figure 3p), which may be due to lower spatial resolution than the region- and basin-scale models used here (Table B2), the computational tradeoff of it being a global simulation. HYCOM25 nevertheless has higher values than FESOM-GS and ORCA36. The difference we

find between simulations tidally forced and not is consistent with previous studies which argue that in order to emulate the upcoming SWOT observations, applying tidal forcing is a key aspect in addition to model resolution (Savage et al., 2017a, b; Arbic et al., 2018; Torres et al., 2018; Ajayi et al., 2021; Yu et al., 2021). The benefit of having tidally forced simulations is that we can develop and test such methods of removing tides.

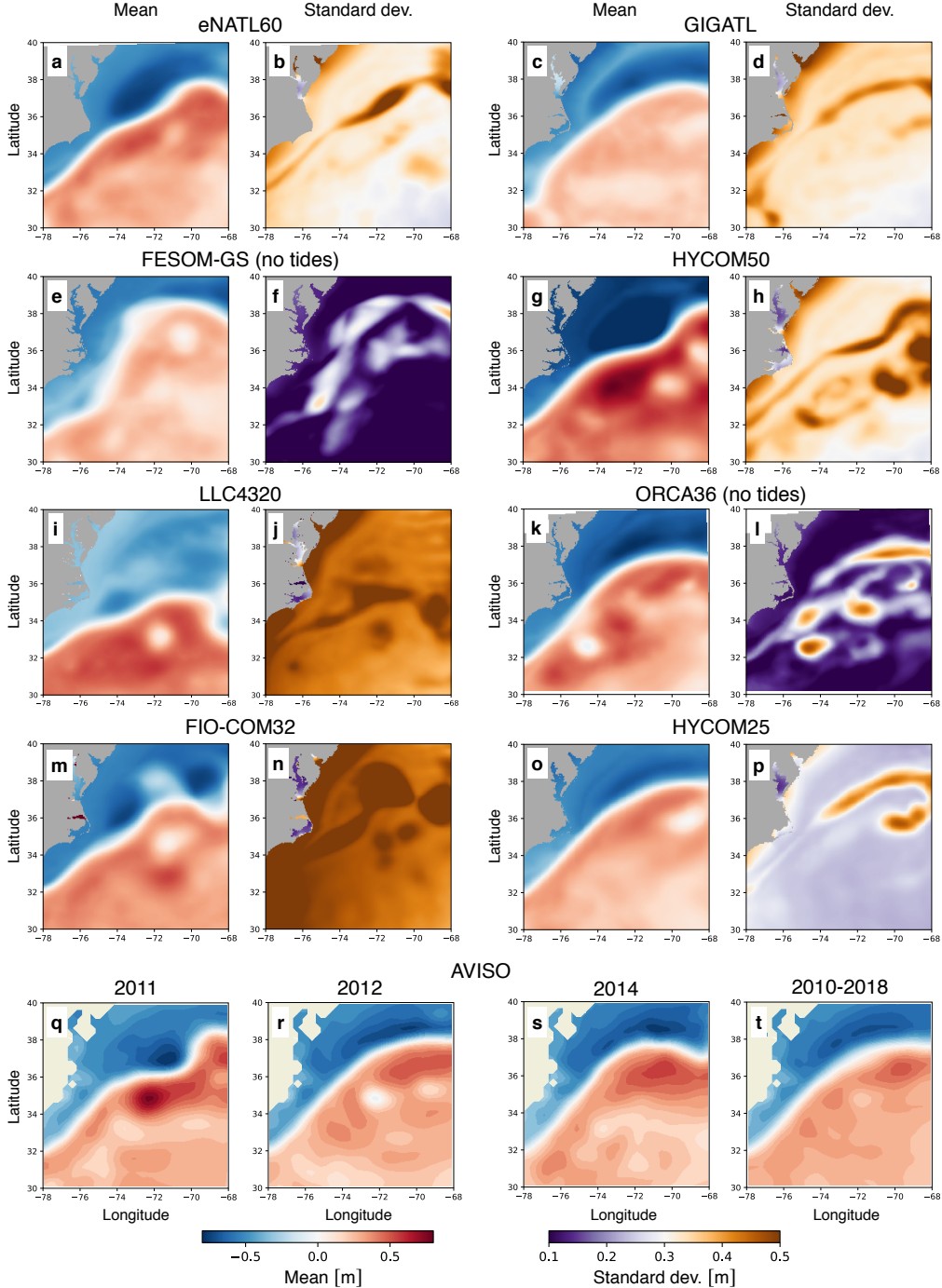

**Figure 3.** The temporal mean and standard deviation of ocean dynamic sea level in the Gulf Stream separation region (Region 1) during the months of February, March and April using hourly outputs of SSH from the models. The bottom row shows the seasonal mean of ADT fields from AVISO during the months of February, March and April. Daily AVISO data were used to compute the seasonal mean for three individual years (2011, 2012 and 2014) and over 2010-2018. The spatial mean is subtracted from the temporal mean fields from the models and AVISO to ensure that the mean SSH/ADT anomaly fields are comparable (*i.e.*, large-scale contributions have been removed).

To complement the temporal standard deviation, in Figure 4, we show the frequency spectra of SSH in the GS separation region. The frequency periodograms were computed every $\sim 10\,\mathrm{km}$ using the `xrft` Python package (Uchida et al., 2021) and then spatially averaged to compute the spectra. The temporal linear trend was removed and a Hann window was applied prior to taking the Fourier transform of SSH as commonly done in studies examining spectra (*e.g.*, Uchida et al., 2017; Savage et al., 2017a; Khatri et al., 2021). At frequencies higher than the Coriolis frequency, LLC4320 shows the highest variability and FESOM-GS the lowest for both winter and summer. FIO-COM32 shows the largest spectral amplitudes at the diurnal and semi-diurnal frequencies amongst the models, which reflects itself in the large standard deviation (Figure 3n). LLC4320 also shows the largest number of spectral peaks at tidal frequencies, likely due to being forced with full lunisolar tidal potential as opposed to a discrete number of tidal constituents, as was the case for the other models used here (Table B6). Also note that tidal forcing in the LLC4320 simulation was inadvertently overestimated by a factor of 1.1121. It is not surprising that FESOM-GS lacks spectral peaks at diurnal and semidiurnal frequencies, considering that it is not tidally forced. ORCA36, on the other hand, although not tidally forced displays some activity at diurnal and semidiurnal frequencies possibly due to the inclusion of atmospheric pressure variation in their forcing (Table B6). However, the lower peaks at tidal frequencies in ORCA36 compared to the tidally forced runs reflect themselves in the lower standard deviation as seen in Figure 3l. eNATL60, GIGATL, HYCOM50 and HYCOM25 show similar levels of variability in the diurnal and semidiurnal band. It is interesting to note that at time scales of $O(1\text{-}10\ \mathrm{days})$, most runs show higher variability during winter than summer (Figure 4a,c), while the tidally forced runs show higher variability at time scales shorter than $O(1\ \mathrm{day})$ during summer (Figure 4b,d). The seasonality at time scales shorter than $O(1\ \mathrm{day})$ is reversed for ORCA36, a run with no tidal forcing. It is possible that the increase in forward cascade of energy stimulated by the tides are the culprit for higher SSH variability at time scales shorter than the inertial frequency during summer than winter for the tidally forced runs and vice versa for the non-tidally forced runs (Barkan et al., 2021). The overall higher SSH variability at time scales longer than the inertial frequency during winter than summer, on the other hand, is possibly due to wind-driven inertial waves (Flexas et al., 2019).

## 3.2  Three-dimensional diagnostics on physical processes

To exemplify 3D diagnostics, we display the submesoscale vertical buoyancy flux from each model using the daily-averaged outputs. Submesoscale vertical buoyancy fluxes in the surface ocean have been of great interest to the ocean and climate modeling community as they modulate the air-sea heat flux, affect mixed-layer depth (MLD), and are a proxy for baroclinic instability taking place within the mixed layer (often referred to as mixed-layer instability (MLI); Boccaletti et al., 2007; Mensa et al., 2013; Johnson et al., 2016; Su et al., 2018; Uchida et al., 2017, 2019; Schubert et al., 2020; Khatri et al., 2021). Ocean models used for climate simulations, however, lack the spatial resolution to resolve MLI due to computational constraints. A recent parametrization proposed by Fox-Kemper et al. (2008) has been operationally implemented by multiple climate modeling groups (Fox-Kemper et al., 2011; Huang et al., 2014; Calvert et al., 2020). While the vertical buoyancy flux predicted by the MLI parametrization has been tested in idealized simulations (Boccaletti et al., 2007; Fox-Kemper and Ferrari, 2008; Brannigan et al., 2017; Callies and Ferrari, 2018), non-eddying and mesoscale-permitting coupled and ocean-only simulations (Fox-Kemper et al., 2011; Calvert et al., 2020), and single-model assessments (*e.g.*, Mensa et al., 2013; Li

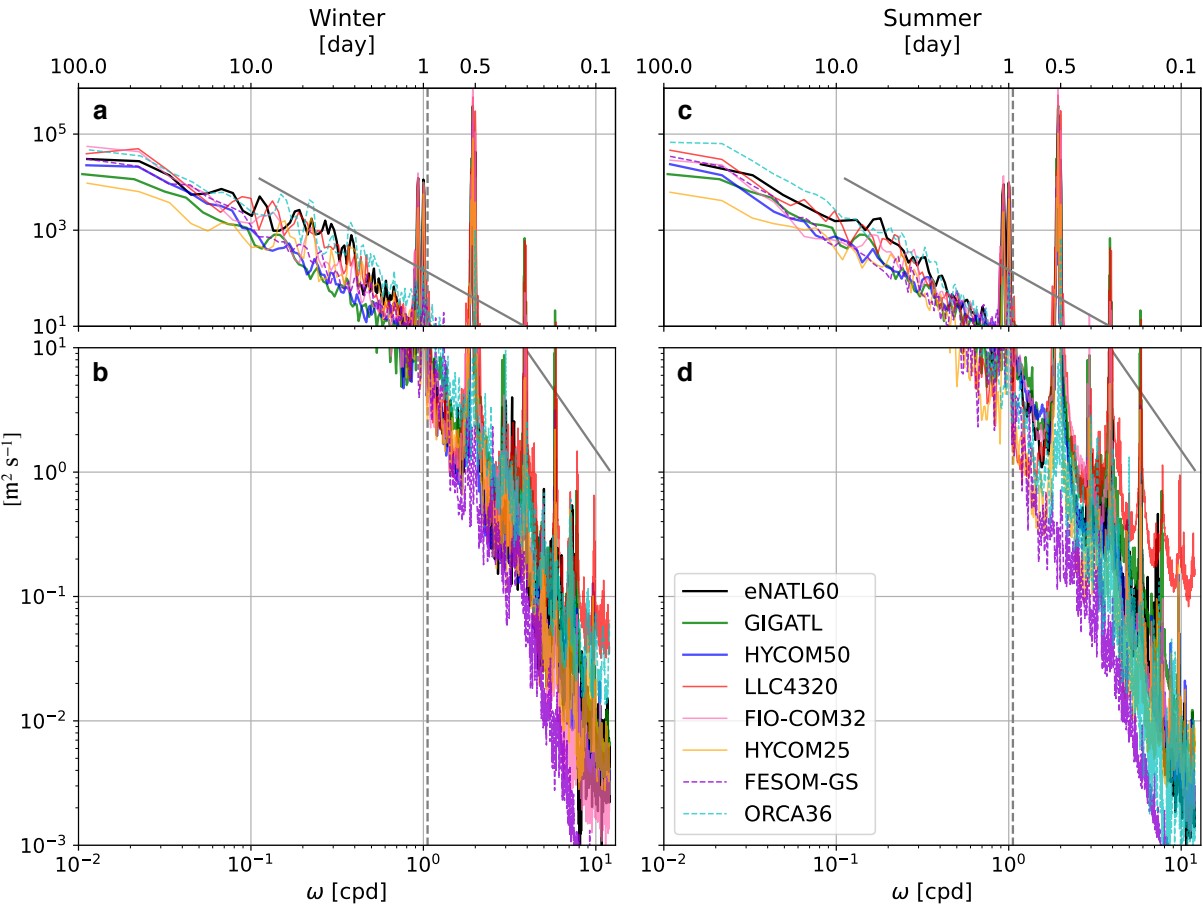

**Figure 4.** The frequency spectra of hourly SSH for winter (February, March, April; left) and summer (August, September, October; right). The panels are split up at $10\,\mathrm{m^2\,s^{-1}}$ for visualization purposes. The frequency periodograms were computed every $\sim 10\,\mathrm{km}$ in Region 1 and then spatially averaged. The runs without tidal forcing (FESOM-GS and ORCA36) are shown in dashed lines. The Garrett-Munk spectral slope of $\omega^{-2}$ (Garrett and Munk, 1975) is shown as the grey solid line and the domain-averaged Coriolis frequency as the grey dashed line.

et al., 2019; Yang et al., 2021; Richards et al., 2021), to our knowledge, a systematic assessment of the MLI parametrization has not been done versus multi-model, submesoscale-permitting, realistic ocean simulations which at least partially resolve the flux in need of parametrization in climate simulations. We take advantage of the unique opportunity provided by our collection of simulations to assess the flux parametrization, *i.e.*, the covariance of the 3D vertical velocity and buoyancy fields versus the modeled mixed layer depth and horizontal buoyancy gradient (3D data were not available for the HYCOM25 simulation).

The MLI parametrization predicts that the submesoscale vertical buoyancy fluxes vertically averaged over the mixed layer $(\overline{(\cdot)}^z)$ can be approximated by the squared horizontal gradient of the mesoscale buoyancy field times the mixed layer depth squared:

$$\overline{w^s b^s}^z \propto \frac{H_{\mathrm{ML}}^2 \overline{|\nabla_{\mathrm{h}} b^m|}^{z\,2}}{|f|}, \tag{1}$$

where $w$, $b$, $f$ and $H_{\mathrm{ML}}$ are the vertical velocity, buoyancy, local Coriolis parameter and MLD. While each model used a different Boussinesq reference density ($\rho_0$), buoyancy was defined as $b = -g\frac{\sigma_0}{\rho_0}$ where $\sigma_0$ is the potential density anomaly with the reference pressure of $0\,$dbar and $\rho_0 = 1000\,\mathrm{kg\,m}^{-3}$ for all model outputs. The MLD was defined using the density criterion (de Boyer Montégut et al., 2004), viz. the depth at which $\sigma_0$ increased by $0.03\,\mathrm{kg\,m}^{-3}$ from its value at $\sim 10\,\mathrm{m}$ depth. $\nabla_{\mathrm{h}}$ is the horizontal gradient and the superscripts $s$ and $m$ indicate the submeso- and meso-scale field respectively. The decomposition between the submeso- and meso-scale were done by applying a Gaussian filter with the standard deviation of $30\,$km using the `gcm-filters` Python package (Grooms et al., 2021). Namely, the mesoscale field is defined as the spatially smoothed field with the Gaussian filter and submesoscale as the residual $(\cdot)^s = (\cdot) - (\cdot)^m$. The $b^m$ field includes scales larger than the typical mesoscale but as it is the horizontal gradient of this field we are interested in, $\nabla_{\mathrm{h}} b^m$ captures the mesoscale fronts. We note that the Gaussian filter, implemented as a diffusive operator, commutes with the spatial derivative (this is an important property as we take the horizontal gradient of $b^m$; Grooms et al., 2021). While we acknowledge that there may be more sophisticated methods to decompose the flow (Uchida et al., 2019; Jing et al., 2020; Yang et al., 2021), a spatial filter has been commonly applied in examining the submesoscale flow in realistic simulations (*e.g.*, Mensa et al., 2013; Su et al., 2018; Li et al., 2019; Jing et al., 2021). Recently, Cao et al. (2021) argued that in addition to spatial cutoffs, a temporal cutoff improves the decomposition. Upon examining the frequency-wavenumber spectra of relative vorticity and horizontal divergence, however, we found that the daily averaging effectively filtered out the internal gravity waves (not shown). Based on characteristic time scale arguments, it is likely that our daily-averaged submesoscale fields are capturing the component in balance with stratification and Earth's rotation (Boccaletti et al., 2007; McWilliams, 2016), although some of the submesoscale balanced variability and nearly all of the internal gravity wave variability is filtered out by the daily average. Figure 5 shows the decomposition for $w$ and $b$ from eNATL60 on February 1, 2010 at depth $18\,$m. We see the characteristic feature of the Gulf Stream separation particularly in the buoyancy field (Figure 5d) and submesoscale fronts (Figure 5c,f) superimposed on top of the large scale flow (Figure 5b,e). We will focus on the late winter/early spring months (February, March and April) as the spatial scale of MLI during summer is not well resolved even at kilometric resolution (Dong et al., 2020). We also restrict our diagnostics to the open ocean where the bathymetry is deeper than $100\,$m (*e.g.*, Figure 7).

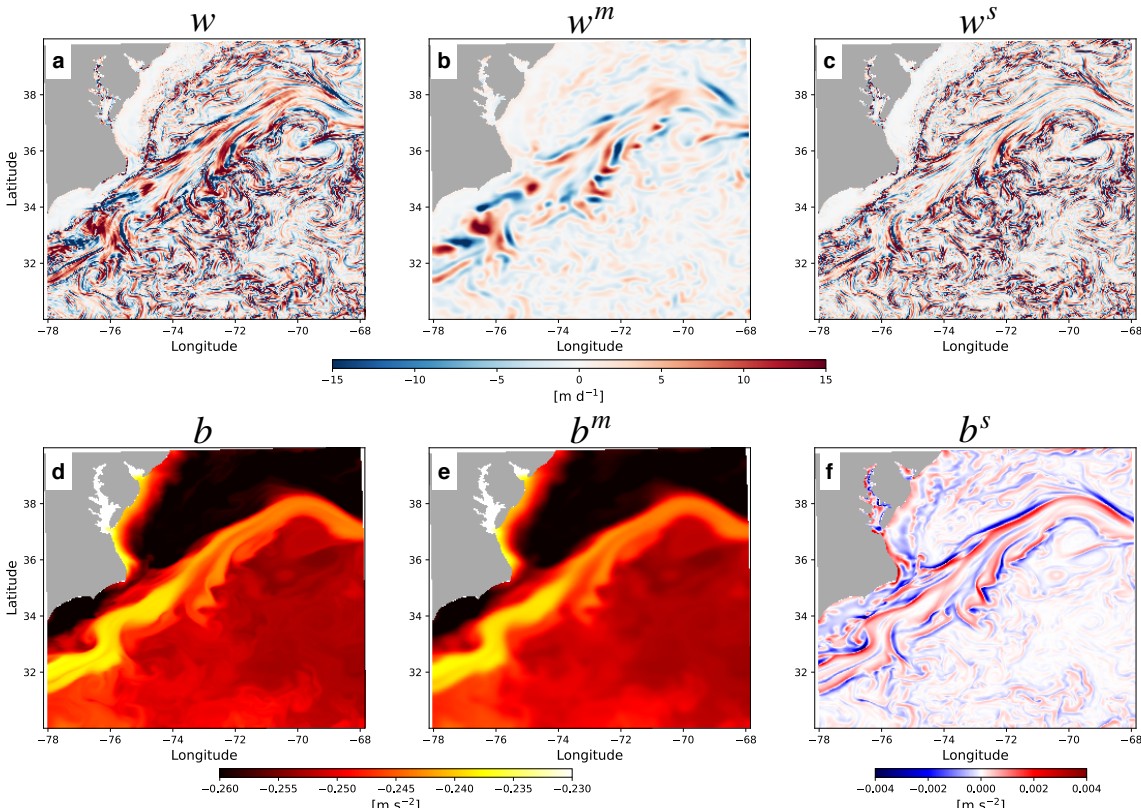

**Figure 5.** Snapshot from eNATL60 on February 1, 2010 at depth 18 m of the unfiltered daily $w$ and $b$ (left column), filtered fields applying the Gaussian filter ($w^m, b^m$; middle column), and the residual ($w^s, b^s$; right column).

Considering that the Fox-Kemper et al. (2008) MLI parametrization is intended for mesoscale-permitting models (neglecting the dependency on model grid-scale: $\Delta s$ in Fox-Kemper et al., 2011), we further coarse grained the fields to $\sim 1/12°$ with a box-car operator, which gives:

$$\langle \overline{w^s b^s}^z \rangle \simeq C_e |f|^{-1} \left( \int_{-\langle H_{\mathrm{ML}} \rangle}^{0} \langle |\nabla_{\mathrm{h}} b^m| \rangle dz \right)^2, \tag{2}$$

where $\langle \cdot \rangle$ is the coarse-graining operator and $C_e$ a tuning parameter or 'efficiency coefficient' (Fox-Kemper et al., 2011). The $\Delta s$ scaling to compensate for coarse model resolution was omitted due to all our model outputs partially resolving the submesoscale buoyancy flux. Furthermore, as $\Delta s$ doesn't vary much among the models, this factor would not contribute much to the overall differences between models, in comparison to the greater variability due to numerics, *etc.*, this manuscript is meant to introduce. We diagnosed $C_e$ by taking the ratio between the right-hand and left-hand side of equation (2) at each grid

point and time step (*e.g.*, left columns of Figure E1), and then the horizontal spatial median of it. The diagnosis (2) would differ from the parametrization (1) if there are large vertical variations in the buoyancy gradient, but these are not expected within the

frequently-remixed mixed layer. Furthermore, the efficiency coefficient is expected to vary among the multi-model ensemble according to how well-resolved and/or damped the submesoscale instabilities are by model numerics, sub-grid schemes, and daily averaging.

The diagnosed $C_e$ only has a time dependence and fluctuates between the range of $[0.01, 0.07]$ across most models (blue solid curves in Figure 6) in agreement with the value of $0.06$ recommended by Fox-Kemper et al. (2008). The time series of the spatial median of $\langle \overline{w^s b^s}^z \rangle$ and its prediction from the MLI parametrization are in sync with each other (black and red solid curves in Figure 6). The order of magnitude of the spatial median of the submesoscale vertical buoyancy flux diagnosed from the models ($O(5 \times 10^{-9}\,\mathrm{m}^2\,\mathrm{s}^{-3})$) also agrees with observational estimates (Mahadevan et al., 2012; Johnson et al., 2016;

Buckingham et al., 2019) with an overall decrease in amplitude towards May except for FIO-COM32, which shows a local maximum around March (black solid curves in Figure 6).

     We provide a snapshot of $\langle \overline{w^s b^s}^z \rangle$ and its prediction from the MLI parametrization (*i.e.*, both sides of equation (2)) on February 1 from each model in Figure 7. The joint histograms of the two over the months of February-April are also given in the bottom rows of Figure 7. The joint histograms of the two are concentrated around the one-to-one line indicating spatial

correlation. The slight underestimation of magnitude in the MLI parametrization (viz. values falling below the one-to-one line) comes from the fact that while $\langle \overline{w^s b^s}^z \rangle$ can take negative values locally where frontogenesis dominates (*i.e.*, where the isopycnals steepen), the MLI parametrization by construction cannot differentiate between frontogenesis and frontolysis giving only positive values (equation 1). Nonetheless, $\langle \overline{w^s b^s}^z \rangle$ largely takes positive values indicating that processes such as mixed-layer and symmetric instabilities, which yield positive vertical buoyancy fluxes consistent with the extraction of potential

energy (Dong et al., 2021), dominate in the surface boundary layer. While we have taken the spatial median to diagnose $C_e(t)$, which yields the best agreement in the time series (Figure 6), one may decide to instead take the spatial mean or mode, which we discuss in Appendix E.

     For operational purposes, we would like to have a tuning parameter that is independent of not only space but also time. Therefore, we also display the MLI prediction when $C_e$ is a constant taken to be its time mean. The agreement between

260 $\langle \overline{w^s b^s}^z \rangle$ and the prediction remains surprisingly good (red dashed curves in Figure 6); in other words, the MLI parametrization is relatively insensitive to the temporal variability of $C_e(t)$. Regarding inter-model differences, HYCOM50 and LLC4320 have the smallest buoyancy fluxes predicted by the MLI parametrization (*i.e.*, weaker horizontal gradient magnitude and/or shallower mixed layer depths). The smaller predicted values presents itself as $C_e$ diagnosed from the two taking an order of magnitude larger values than the other models (blue curves in Figure 6c,e); particularly for HYCOM50, using a constant $C_e$ fails to

265 reproduce the magnitude of $\langle \overline{w^s b^s}^z \rangle$ during the early half of February (red dashed curve in Figure 6c). It is possible that the lowest vertical resolution of HYCOM50 amongst the models (Table B2) results in under-representing the MLD despite its fine horizontal resolution particularly south of the Gulf Stream (Figure D1c); the MLI parametrization depends on it quadratically (equation (1)). The MLD from LLC4320 is also relatively shallow (Figure D1e). HYCOM50 and LLC4320 both use the K-profile parametrization (KPP, Table B3; Large et al., 1994) for the boundary-layer closure, which may imply that the KPP

parameters warrant further tuning or reformulation for submesoscale-permitting model resolutions (*e.g.*, Bachman et al., 2017;

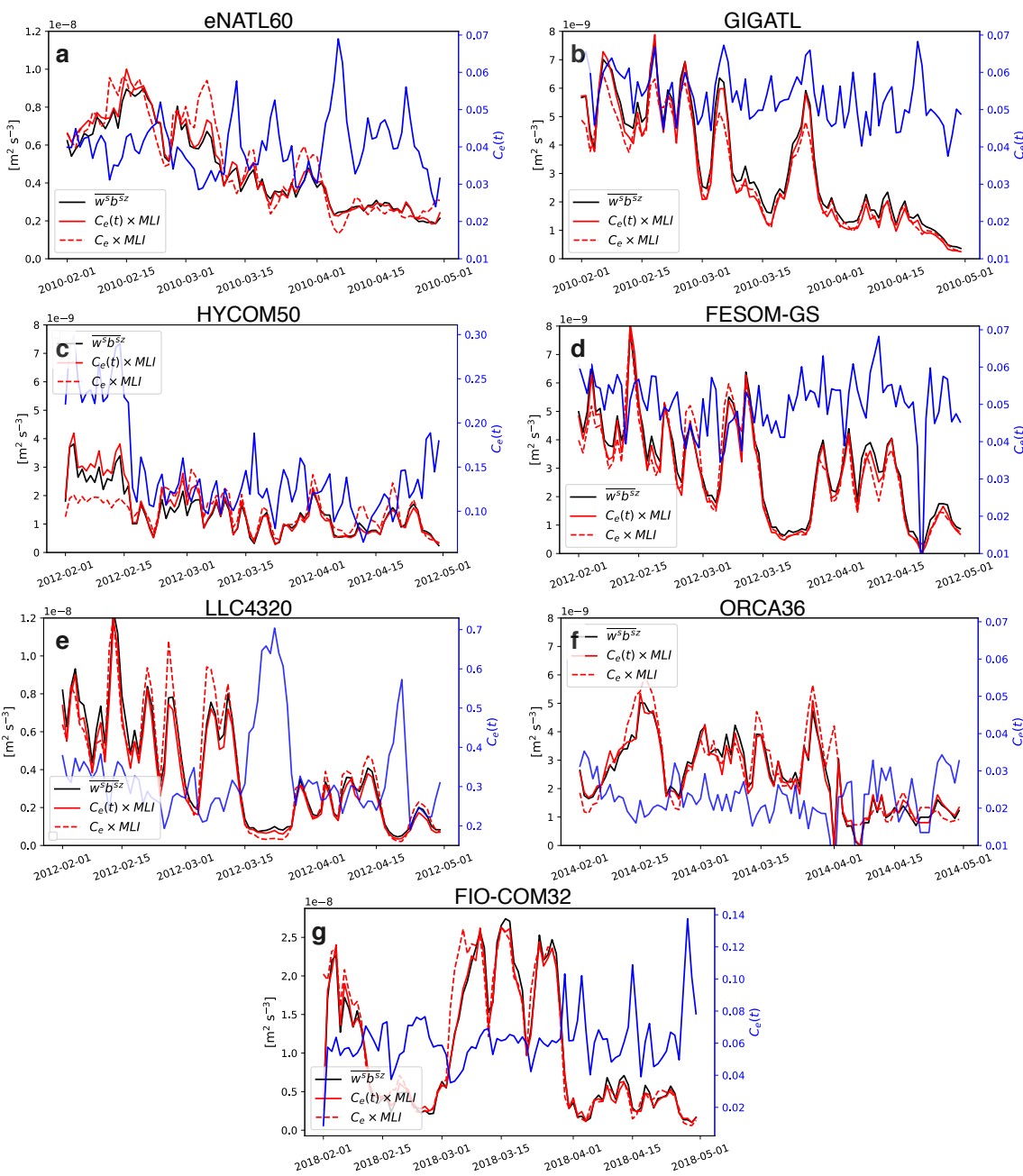

**Figure 6.** Time series of the spatial median of the submesoscale vertical buoyancy flux averaged over the MLD ($\langle\overline{w^s b^s}^z\rangle$; black solid curve) and its prediction from the MLI parametrization during the months of February to April. Note that the $y$ axes vary depending on the magnitude diagnosed from each simulation in order to highlight its temporal variability. The prediction with temporally varying $C_e(t)$ is shown in red solid curves and with a temporally averaged (constant) $C_e$ in red dashed curves. $C_e(t)$ is plotted against the right $y$ axes in blue. Three-dimensional data were not available for HYCOM25.

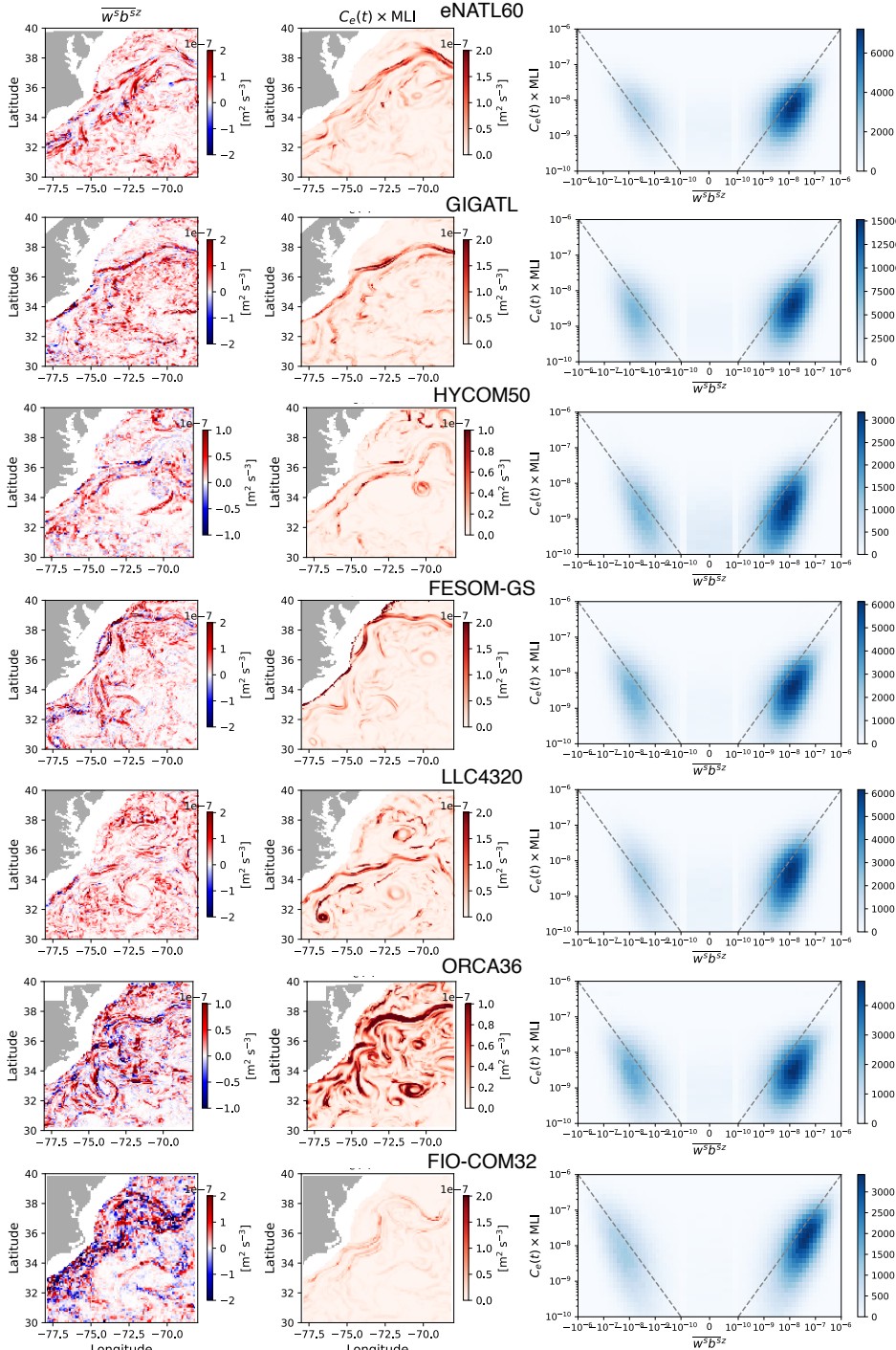

**Figure 7.** Snapshot of $\langle\overline{w^s b^s}^z\rangle$ (left column) and $C_e(t)\times$MLI (middle column) on February 1 for each model. Note that the range of colorbar differs depending on the magnitude diagnosed from each model to highlight their spatial features and comparison between the submesoscale buoyancy flux and its equivalent predicted from the parametrization per simulation. Regions with bathymetry shallower than 100 m are masked out. The right column for each model shows the joint histogram of the two during the months of February to April, and the one-to-one line is shown as the grey dashed line. The histograms were computed using the `xhistogram` Python package (Abernathey et al., 2021c).

Souza et al., 2020). The shallow MLD may also be due to the differences in the atmospheric products used to force the models (Table B6).

## 4    Conditions for sustainability

The strength of cloud storage and computing comes from it being decentralized from any specific institution, but this also leaves open the question about who pays for the cost of operating and supporting the cloud infrastructure, as well as for the cloud resources. There are three components to the cost structure: i) the cloud storage, ii) computation including egress charges to access the data, and iii) deployment and maintenance of the JupyterHub.

Currently as of writing, the operational cost of fluxing data to the OSN cloud storage is funded by an NSF grant acquired by the Climate Data Science Laboratory at Columbia University, and the JupyterHub on Google Cloud Platform (GCP) by a Centre National d'Études Spatiales (CNES) grant acquired by the MultiscalE Ocean Modeling (MEOM) group at the Institut des Géosciences de l'Environnement. While the OSN storage itself allocated to Pangeo Forge is not associated with monetary expense nor any egress charges (https://www.openstoragenetwork.org/get-involved/get-an-allocation/), the scratch storage on GCP where we have saved our diagnostic outputs entailed storage and egress fees. The cost of GCP resources for the Jupyter-Hub with parallelized computation added up to roughly 1000 € per month for this study with the maximum computational resources of 64 cores and 256 gigabytes of memory per user; the resources scale on-demand. As of writing, we have consumed 3.5 tera hours of CPU and 92.1 Tb of RAM monthly on average. The operational scale of 1000 €-per-month worth of GCP credits including the scratch storage and egress charges was too small for us to directly contract with Google so we have gone through a private broker in acquiring the GCP contract. (Our contract with the broker was an additional 600 € per month.) The cost of operating the scalable Kubernetes infrastructure is managed by a non-profit vendor (2i2c) for an additional few thousand dollars a month (https://docs.2i2c.org/en/latest/about/pricing/index.html; We note that the operational cost somewhat depends on the contract negotiated amongst the party of interest, cloud vendor and 2i2c.)

Although the net cost of few thousand euros per month may seem expensive compared to the local download framework where the costs of computation are shouldered upfront upon purchase of the cluster, there are several benefits to a cloud-based approach. First, using cloud infrastructure shifts the burden of hardware maintenance to the cloud provider, and users benefit from regular updates to technology and services that are available, meaning the scientific community can benefit from industry-driven innovations. Second, cloud infrastructure can be managed remotely and may use an inter-operable stack based on standards that are supported by many cloud providers (such as open source tools like Kubernetes and JupyterHub). As all of the underlying technology is open source, it is technically possible for us to have deployed the Kubernetes infrastructure on our own. However, as Earth Scientists, we often do not possess the adequate software engineering skills, and such expertise is highly sought by industry; the hiring of a software engineer at public and higher-educational institutions is difficult due to financial constraints. The service provided by 2i2c makes it easier to port workflows between clouds and get more cost-effective support in operating this infrastructure compared with paying full-time software engineers that run local hardware for an institution.

We would like to note that while we have chosen GCP and OSN for the cloud platform, the core design principles and
technology behind Pangeo Forge and the JupyterHub operated by 2i2c are non-proprietary and cloud vendor agnostic (for
example, as defined in 2i2c's "Right to Replicate"; https://2i2c.org/right-to-replicate). We could re-deploy the entire cloud
platform on a different cloud provider with relative ease. This lets the users of this platform benefit from the flexibility and
efficiency of the cloud, while minimizing the risk of lock-in and dependence on proprietary technology. As the cloud-based
framework spreads within the scientific community, it is also possible that the ocean and climate science community will be
able to negotiate better deals with cloud service providers; the framework is apt for Ocean and Climate Model Intercomparison
Project (OMIP and CMIP; Griffies et al., 2016; Eyring et al., 2016) studies where tera- and peta-bytes of data need to be
shared and analyzed consistently. The systematic storage of ARCO data with open access and appropriate cataloguing will also
enable reproducible science, a crucial step when evaluating newer simulations against previous runs. While we believe we have
provided a proof of concept that cloud computing can be implemented with open-source technologies and can be leveraged
for scientific research, the success of the framework will depend on the scientific community to convince its peers and funding
organizations to recognize its benefit.

## 5  Conclusions

In this study, we have implemented a cloud-based framework for collaborative, open-source and reproducible science, and have
showcased its potential by analyzing eight submesoscale permitting simulations at a SWOT Crossover (Xover) region around
the Gulf Stream separation (Region 1 in Figure 1). We have shown that despite the similar horizontal resolution amongst many
models in this study, the spatial scales represented vary widely (Figure 2). This diverse representation likely originates from
differences in advective/diffusive schemes, boundary layer parametrizations, atmospheric and tidal forcing, vertical resolution
and/or bathymetry, and potentially duration of spin up amongst the simulations used here (Appendix B; cf. Chassignet and Xu,
2021). The need for collaborative work to inter-compare realistic simulations stems from both a scientific interest in the fidelity
of submesoscale-permitting ocean models in representing the underlying physics and tracer transport, and an engineering
perspective on the numerics of ocean models. We leave a detailed analysis on the impact of numerics on the resolved dynamics
for future work.

We have provided example diagnostics on SSH variability and submesoscale vertical buoyancy fluxes. The temporal standard
deviation and spectra of SSH were significantly lower for the simulations without tidal forcing compared to the tidally forced
simulations (Figures 3 and 4). This implies that in order to emulate the upcoming SWOT altimetric observations, tidal forcing
is a key factor in modeling the surface ocean (Savage et al., 2017a, b; Arbic et al., 2018; Yu et al., 2021; Barkan et al.,
2021; Le Guillou et al., 2021). Regarding 3D diagnostics, both the good agreement across multiple models between the tuning
parameter $C_e$ in the MLI parametrization and the values recommended by its developers (Figure 6; Fox-Kemper et al., 2008;
Fox-Kemper and Ferrari, 2008), and the consistency of the order of magnitude of the flux predicted by the parametrization
in the spatially averaged sense with observational estimates (cf. Richards et al., 2021), combine to provide confidence in
implementing the MLI parametrization in realistic ocean and climate models. This is in contrast, however, with a recent study

by Yang et al. (2021, their Figure 7) where they found (using the Regional Ocean Modeling System, ROMS with KPP) that the time series of $\langle \overline{w^s b^s}^z \rangle$ did not correlate well with the prediction from its parametrization in the Kuroshio extension. While we lack access to their model outputs, we speculate that the differences could be due to the diagnostic methods, domain of interest and/or configuration of their simulation. The contrasting findings all the more highlight the need for collaborative and open data analysis strategies of multi-model ensembles in assessing and improving the simulations themselves. We would like to note that were the modeled domain by Yang et al. (2021) covered Region 1, the cloud-based framework would allow for a straightforward platform to extend the ensemble of simulations (Appendix B) to include their outputs for our inter-comparison and reproducible science.

We end by noting that cloud-based data-proximate computation provides a framework to systematically analyze tera- and peta-bytes of data as we further increase the resolution and complexity of ocean and climate simulations, and as SWOT data becomes available. However, the success of the framework will depend on the ability of scientists to convince funding organizations to recognize its potential. Cloud-based computing differs from the conventional workflow which involves funding local computational resources and storage. While the cloud-based framework does not allow for an individual researcher or group to have prioritized access over the data and analytical tools, we believe that open access to the data will allow for reproducible science and facilitate international collaboration.

*Code and data availability.*  The model outputs from eNATL60, GIGATL, HYCOM50, FESOM-GS, ORCA36, FIO-COM32 and HYCOM25 at the SWOT-Xover regions are all publicly available on the Open Storage Network (OSN). The Jupyter notebooks and Yaml file used to access and analyze the data are available on Github (https://github.com/roxyboy/swot_adac_ogcms/tree/notebook; a DOI will be added upon acceptance of the manuscript). The LLC4320 data were accessed via the NASA ECCO Data Portal (https://data.nas.nasa.gov/ecco/data.php?dir=/eccodata/llc_4320) using the `llcreader` of the `xmitgcm` Python package (Abernathey et al., 2021d; Abernathey, 2019).

## Appendix A: Example of `pangeo_forge_recipe` for eNATL60

Here we provide the Pangeo Forge recipe used to flux eNATL60 surface hourly data to OSN for Region 1 during February and April, 2010. The `input_url_pattern` is where the original NetCDF files were hosted on an OPeNDAP server, upon which the files were chunked along the time dimension before being fluxed to the cloud in Zarrified format (Miles et al., 2020). As a contributor to Pangeo Forge, one essentially only needs to specify the `input_url_pattern`. The zarrification and fluxing of the data to the cloud is automated by Pangeo Forge, reducing the infrastructure and cognitive burden on the data provider (Stern et al., 2022).

**Listing 1.** eNATL60 example

```
from itertools import product

import pandas as pd
from pangeo_forge_recipes.patterns import pattern_from_file_sequence
```

```
          from pangeo_forge_recipes.recipes import XarrayZarrRecipe

regions = [1]
          season_months = {
              "fma": pd.date_range("2010-02", "2010-05", freq="M")
          }

url_base = (
              "https://ige-meom-opendap.univ-grenoble-alpes.fr"
              "/thredds/fileServer/meomopendap/extract/SWOT-Adac"
          )

def make_recipe_surface(region, season):
              input_url_pattern = url_base + "/Surface/eNATL60/Region{reg:02d}-surface-hourly_{yymm}.nc"
              months = season_months[season]
              input_urls = [
                  input_url_pattern.format(reg=region, yymm=date.strftime("%Y-%m")) for date in months
]
              file_pattern = pattern_from_file_sequence(input_urls, "time_counter")

              target_chunks = {"time_counter": 72}
              subset_inputs = {"time_counter": 3}
recipe = XarrayZarrRecipe(
                  file_pattern, target_chunks=target_chunks, subset_inputs=subset_inputs
              )
              return recipe

recipes = {
              f"eNATL60/Region{reg:02d}/surface_hourly/{season}": make_recipe_surface(reg, season)
              for reg, season in product(regions, season_months)
          }
```

## Appendix B: Model configurations

We provide the model configurations in Tables B1-B6 (blanks indicate the information was not obtainable). The vertical coordinate transformation onto geopotential coordinates for the outputs of GIGATL and HYCOM50, which had terrain-following and isopycnal coordinates as their native grid respectively (Table B2), were done using the xgcm Python package (Abernathey

et al., 2021b) with linear interpolation. For the sake of storage, only three months of output for summer (Aug., Sep., Oct.) and winter (Feb., Mar., Apr.) respectively are stored on OSN from an arbitrary year per simulation.

## Appendix C: Impact of spatiotemporal smoothing on the temporal standard deviation

In this appendix, we examine the effect of spatiotemporal filtering on the modelled SSH standard deviation. In order to mimic a smoothing procedure similar to the AVISO products, we apply a Gaussian spatial filter with the standard deviation of $50\,\mathrm{km}$ using the `gcm-filters` Python package and a 10 day running mean (cf. Chassignet and Xu, 2017). The non-tidally forced runs do not show much difference upon spatiotemporal smoothing from their standard deviation using hourly outputs but, they significantly decrease for the tidally forced runs, particularly LLC4320 and FIO-COM32, with the modelled amplitudes coming closer to the AVISO estimate (Figures 3 and C1). The strong reduction in LLC4320 and FIO-COM32 may be expected as they are the runs with highest SSH variance at frequencies higher than the Coriolis frequency (Figure 4). All simulations agree that there is a local maximum in standard deviation around $37°\mathrm{N}$ where the separated GS situates consistent with AVISO. The SSH variability in GIGATL may be on the lower end considering it is tidally forced (Figure C1b), which could also be due to the lack of pressure variation in the atmospheric forcing (Table B6).

## Appendix D: Mixed-layer depth

The MLD averaged between February 1–15 is shown in Figure D1 along with the climatology for the month of February estimated from the Argo floats. We see that the MLD from HYCOM50 and LLC4320 are notably shallower south of the Gulf Stream compared to the other models and Argo estimate.

## Appendix E: Efficiency coefficient and the MLI parametrization sensitivity to it

The efficiency coefficient $C_e(t,x,y)$ diagnosed from each simulation is given in the left column of Figure E1 and the joint histogram where $C_e(t)$ is taken as the spatial mean and mode in the right two columns respectively. It is interesting to note that $C_e(t,x,y)$ tends to take small values within fronts (namely, where the magnitude of $\langle \overline{w^s b^s}^z \rangle$ is large), but takes large values, reaching up to $O(1)$, on their periphery (Figures 7 and E1). Comparing the joint histograms in Figures 7 and E1, taking the spatial mean to diagnose $C_e(t)$ tends to overestimate the flux magnitude predicted from the parametrization as the mean is sensitive to extrema than the median; the histograms are concentrated above the one-to-one line (middle column of Figure E1). Diagnosing $C_e(t)$ as the spatial mode seemingly improves the alignment of the histogram with the one-to-one line (right column of Figure E1). However, taking the spatial mode results in $C_e(t)$ reaching values up to two orders of magnitude larger than the values recommended by Fox-Kemper et al. (2008), and the time series predicted from the parametrization results in overestimating the submesoscale buoyancy flux in the spatially averaged sense (Figure E2). The time series predicted from using the spatial mean to estimate $C_e(t)$ further overestimates the buoyancy flux (not shown). We, therefore, recommend the usage of spatial median in estimating $C_e(t)$.

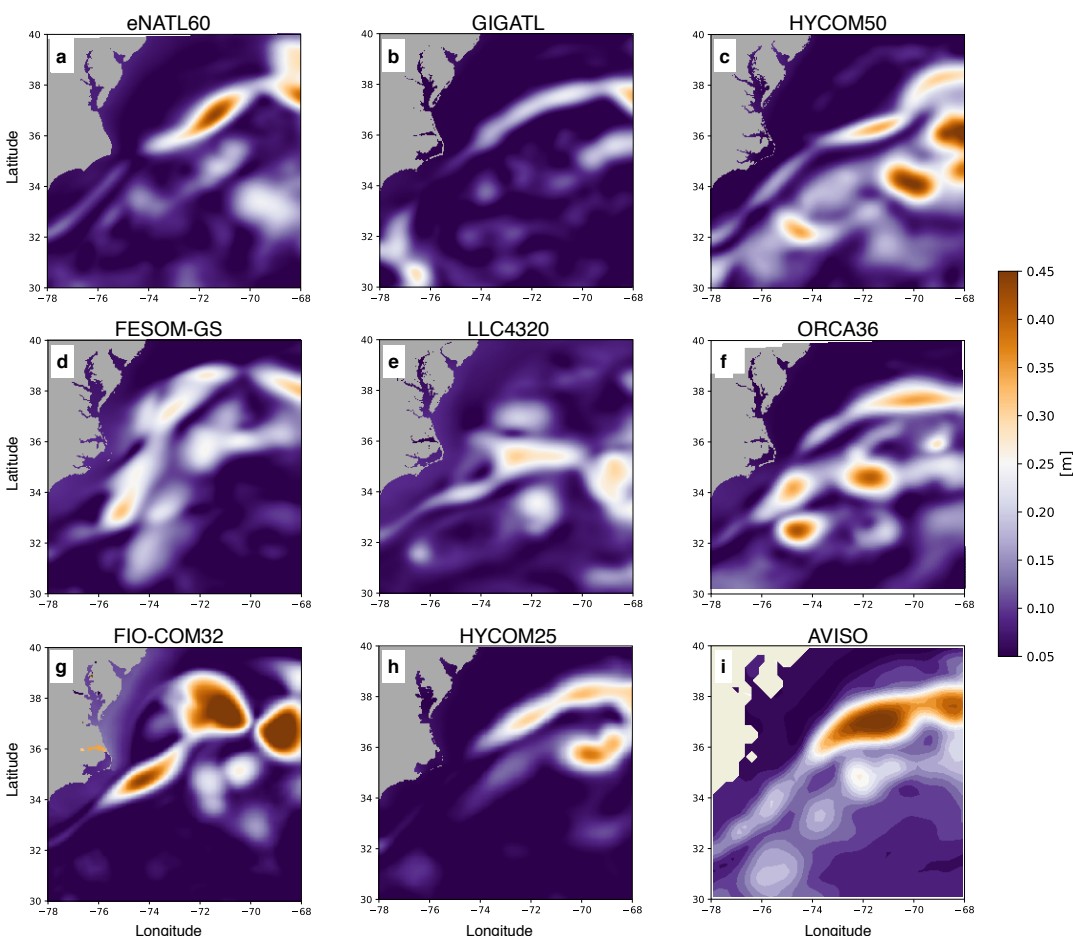

**Figure C1.** Temporal standard deviation of the spatiotemporally smoothed SSH from the eight models and ADT from AVISO over the years of 2010-2018 during the months of February-April. Note that the colorbar is slightly adjusted from Figure 3 in order to accommodate for lower values.

*Author contributions.* Conceptualization, T.U. & J.L.S.; methodology, T.U.; software, T.U., A.A., L.B., C.S., R.P.A. & C.H.; validation, T.U.; formal analysis, T.U.; investigation, T.U. J.L.S., B.F-K. & W.K.D.; computational resources, J.L.S. & R.P.A.; data curation, J.L.S., A.A., L.B., E.P.C., X.X., J.G., G.R., N.K., S.D., Q.W., D.M., C.B., B.K.A., J.F.S., F.Q., B.X., A.B., R.S. & A.W.; writing, T.U.; visualization, T.U.; project administration, J.L.S.; funding acquisition, J.L.S. & R.P.A. All authors have read and agreed to the published version of the manuscript.

*Competing interests.* The authors claim no competing interests.

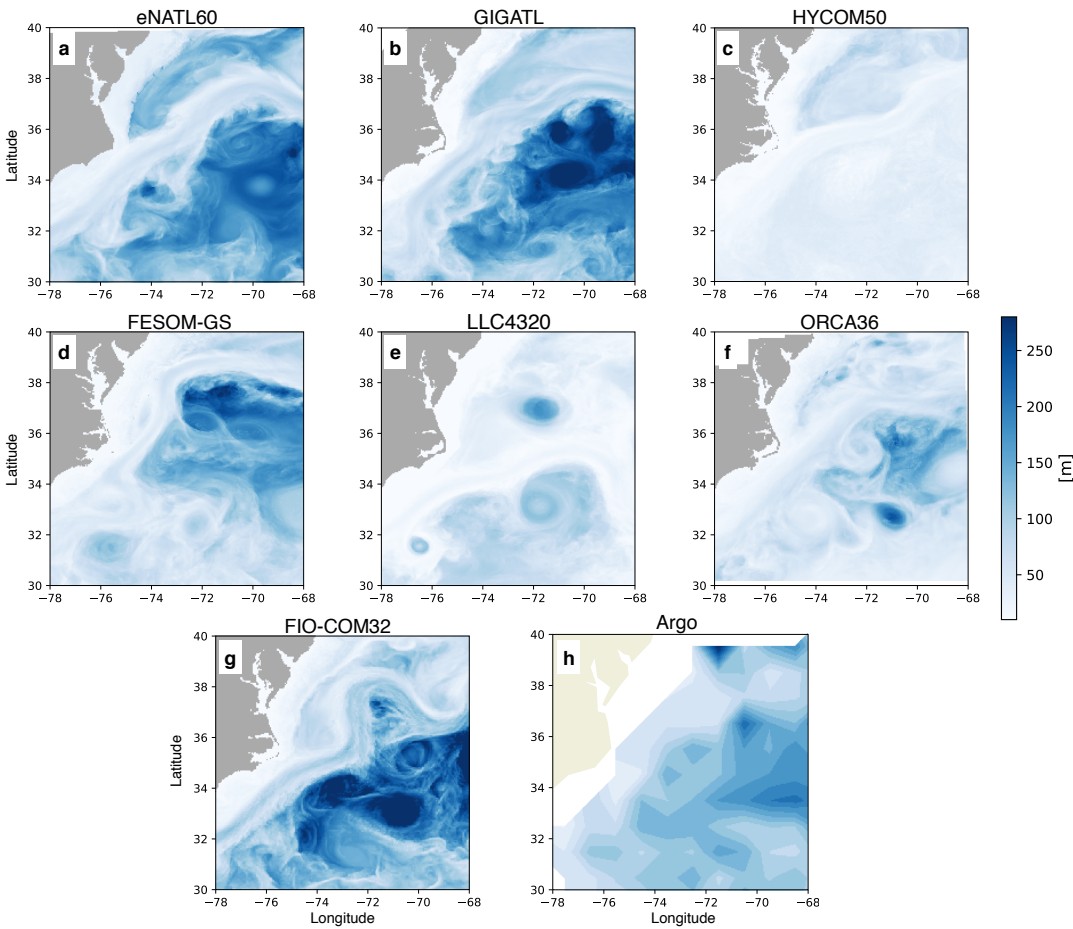

**Figure D1.** MLD from each model averaged over the duration of February 1–15 when the prediction from the MLI parametrization with a constant $C_e$ in HYCOM50 deviates from the diagnosed submesoscale vertical buoyancy flux. The MLD was defined using the density criteria of de Boyer Montégut et al. (2004). For models with non-geopotential vertical coordinates (*i.e.*, GIGATL and HYCOM50), the MLD was computed using their native coordinates respectively. The climatology for the month of February from the Argo floats is taken from the dataset by Holte et al. (2017). The monthly-mean MLD defined by the density criterion (`mld_dt_mean`) is shown in order to be consistent with our model estimates.

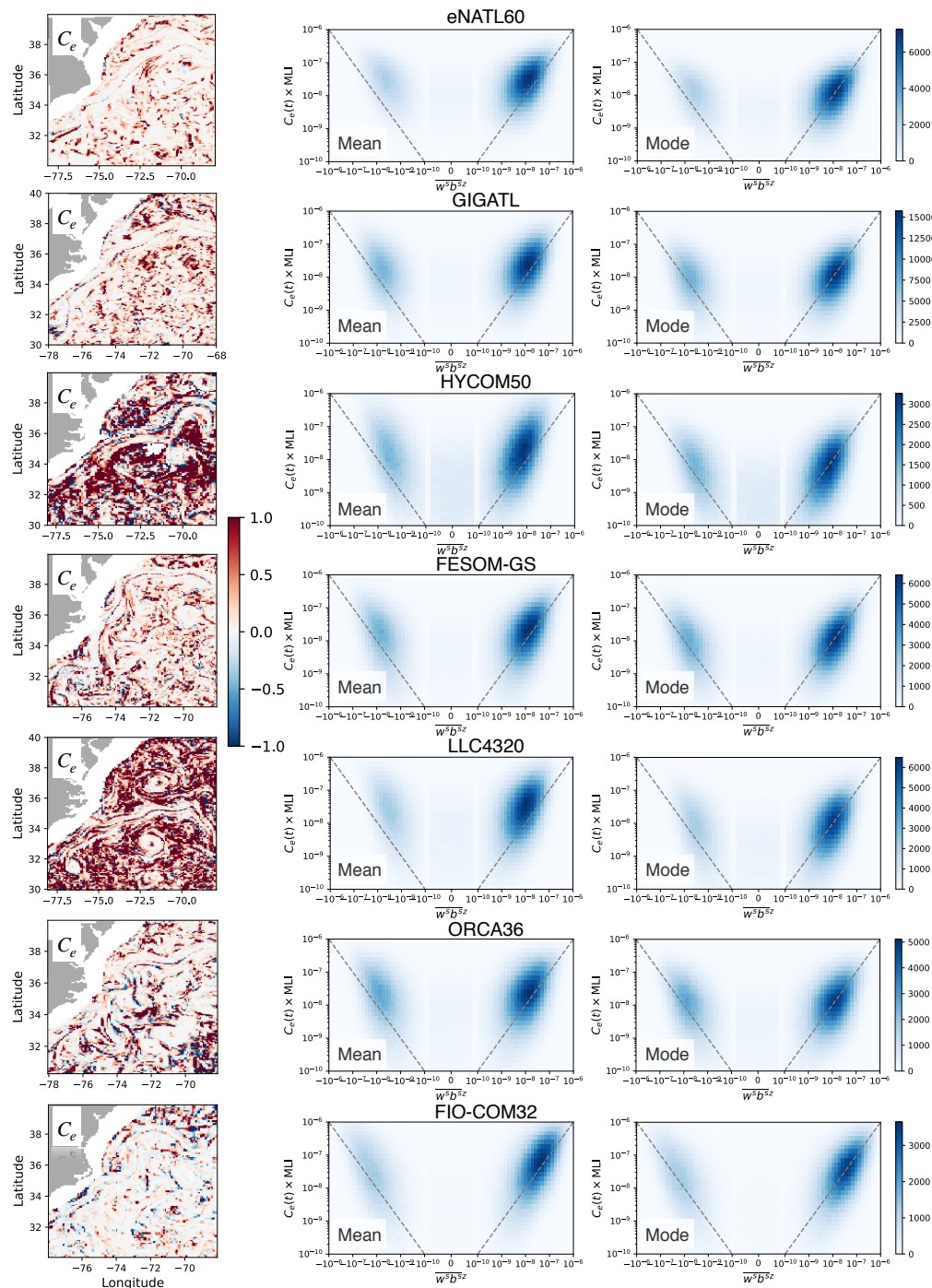

**Figure E1.** Snapshot of the efficiency coefficient $C_e(t,x,y)$ diagnosed on February 1 from each simulation (left column). The joint histogram of $\langle \overline{w^s b^s}^z \rangle$ and $C_e(t) \times$ MLI during the months of February to April (right columns). The middle column shows the histogram when $C_e(t)$ is taken as the spatial mean of $C_e(t,x,y)$ and right column as the spatial mode of $C_e(t,x,y)$. The one-to-one line is shown as the grey dashed line.

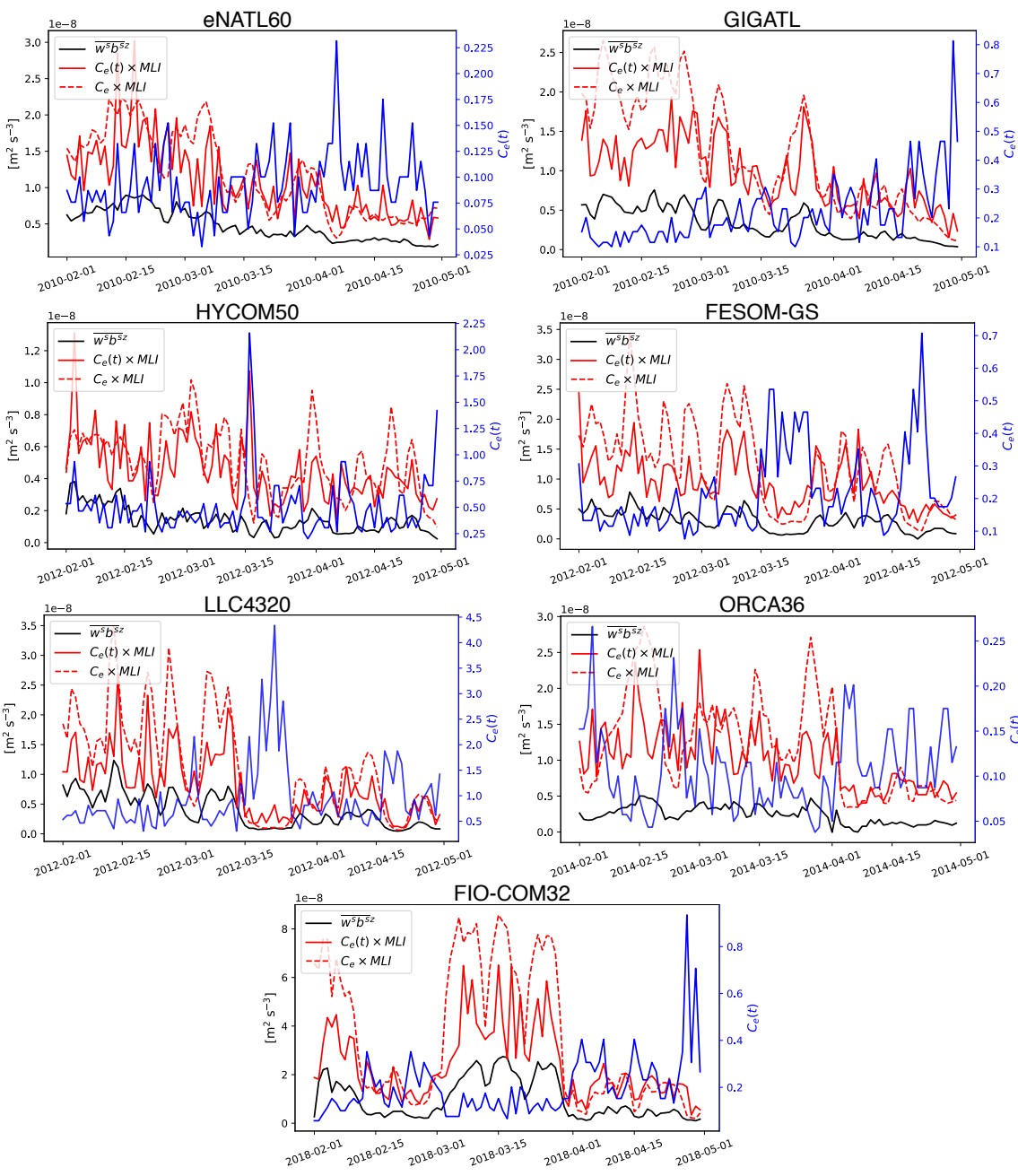

**Figure E2.** Time series of the spatial median of the submesoscale vertical buoyancy flux averaged over the MLD ($\langle\overline{w^s b^s}^z\rangle$; black solid curve) and its prediction from the MLI parametrization during the months of February to April where $C_e(t)$ is taken as the spatial mode of $C_e(t, x, y)$. Note that the $y$ axes vary depending on the magnitude diagnosed from each simulation in order to highlight its temporal variability. The prediction with temporally varying $C_e(t)$ is shown in red solid curves and with a temporally averaged (constant) $C_e$ in red dashed curves. $C_e(t)$ is plotted against the right $y$ axes in blue.

*Acknowledgements.* We thank the editor Riccardo Farneti, and Stephen Griffies, Mike Bell, Andy Hogg and Joel Hirschi for their reviews. T.U. acknowledges support from the French 'Make Our Planet Great Again' (MOPGA) initiative managed by the Agence Nationale de la Recherche under the Programme d'Investissement d'Avenir, with the reference ANR-18-MPGA-0002. This work is a contribution to the Consistent OceaN Turbulence for ClimaTe Simulators (CONTaCTS) project and SWOT mission Science Team (https://www.swot-adac.org/). J.L.S, L.B. and A.A. acknowledge the PRACE 16$^{th}$ call project ReSuMPTiOn (Revealing SubMesoscale Processes and Turbulence in the Ocean, P.I.: L.B.) for awarding access to the MareNostrum supercomputer at the Barcelona Supercomputing Center. Operational costs for the cloud-based JupyterHub were funded by CNES through their participation in the SWOT Science Team. R.P.A. and C.S. acknowledge support from the NSF award 2026932 for the development of Pangeo Forge and OSN storage. This study is a contribution to the project S2: Improved parameterisations and numerics in climate models (S.D.), S1: Diagnosis and Metrics in Climate Models (N.K.) and M5: Reducing spurious diapycnal mixing in ocean models (S.D.) of the Collaborative Research Centre TRR 181 "Energy Transfer in Atmosphere and Ocean" funded by the Deutsche Forschungsgemeinschaft (DFG, German Research Foundation)—project No. 274762653, and the Helmholtz initiative REKLIM (Regional Climate Change; Q.W.). B.K.A. and J.F.S. acknowledge support from NASA grant 80NSSC20K1135. E.P.C and X.X. acknowledge support from ONR grants N00014-19-1-2717 and N00014-20-1-2769. B.F-K. acknowledges support of ONR N00014-17-1-2963 and NOAA NA19OAR4310366. F.Q. and B.X. acknowledge support from the National Natural Science Foundation of China with the grant No. 41821004. C.B. acknowledges support from the EU H2020 projects IMMERSE (grant agreement No. 821926) and ESIWACE2 (grant agreement No. 823988). J.G. gratefully acknowledges support from the French National Agency for Research (ANR) through the project DEEPER (ANR-19-CE01-0002-01). J.G. and G.R. acknowledge PRACE and GENCI for awarding access to HPC resources Joliot-Curie Rome and SKL from GENCI-TGCC (Grants 2020-A0090112051, 2019gch0401 and PRACE project 2018194735) and HPC facilities DATARMOR of "Pôle de Calcul Intensif pour la Mer" at Ifremer Brest France. W.K.D. acknowledges support from NSF grants OCE-182956 and OCE-2023585. D.M. carried out research at the Jet Propulsion Laboratory, California Institute of Technology, under contract with NASA, with support from the Physical Oceanography and Modeling, Analysis, and Prediction Programs. High-end computing was provided by NASA Advanced Supercomputing at Ames Research Center. We would like to thank 2i2c.org (https://2i2c.org/) for deploying and maintaining the JupyterHub on Google Cloud Platform and NASA ECCO team for maintaining the data portal through which the LLC4320 data were accessed. We acknowledge Stack Labs for brokering the GCP contract. The altimeter products were produced by Ssalto/Duacs and distributed by Aviso+, with support from CNES (https://www.aviso.altimetry.fr). The geographic figures were generated using the `Cartopy` Python package (Met Office, 2010 - 2015).

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

**Table B1.** The model and initial condition used for each simulation, their duration of spin up and year of output stored on OSN. HYCOM50 was spun up from rest and integrated for a total of 20 years. Sensitivity experiments were performed starting from year 15 (Chassignet and Xu, 2017, 2021). LLC4320 used progressive spin-up from a 1/6° state estimate (Menemenlis et al., 2008) followed by 1/12° and 1/24° simulations, as detailed in Table D2 of Rocha et al. (2016).

| Simulation | Model (version) | Initial condition | Spin up | Year of output |
|---|---|---|---|---|
| eNATL60 | NEMO (3.6) | GLORYS12 | 18 months | 2010 |
| GIGATL | CROCO | July 2007 from an identical run w/ 3 km resolution | 12 months | 2010 |
| HYCOM50 | HYCOM | GDEM climatology | 15 years | year 19 |
| FESOM-GS | FESOM (2.1) | PHC3.0 | 18 months | 2012 |
| LLC4320 | MITgcm | ECCO CS510 State Estimate | Progressive | 2012 |
| ORCA36 | NEMO (4.0) | WOA 2013 (temperature & salinity) | 18 months | 2014 |
| FIO-COM32 | FIO-COM (v2.0_HR32) | June 2016 from FIO-COM 1/10° operational ocean forecast w/ data assimilation | 18 months | 2018 |
| HYCOM25 | HYCOM | WOA 2013 | | 2014 |

**Table B2.** The horizontal and vertical native coordinate system, spatial resolution and domain coverage for each simulation. The $Z^*$ vertical coordinate is the rescaled geopotential coordinate where the fluctuations of the free surface are taken into account (cf. Griffies et al., 2016). Note that vertical resolution as well as horizontal resolution vary significantly between the models. Outputs from FESOM-GS were interpolated onto a Cartesian grid off-line with a cubic spline.

| Simulation | Grid structure | Resolution | Vertical coordinate | Domain (grid points: zonal×meri.) |
|---|---|---|---|---|
| eNATL60 | C-grid | 1/60° | $Z^*$ (300 levels) | North Atlantic (8354 × 4729) |
| GIGATL | C-grid | 1 km (nominal) | Terrain following (100 levels) | Atlantic (10500×14000) |
| HYCOM50 | C-grid | 1/50° | Hybrid (32 pressure $p$ & isopycnal) | North & Eq. Atlantic (6709×7373) |
| FESOM-GS | Unstructured | 1/2° w/ refinement to 1 km (nominal) in Region 1 | $Z^*$ (70 levels) | Global (3000502 vertices) |
| LLC4320 | C-grid | 1/48° (nominal) | $Z$ (90 levels) | Global (4320×4320×13 LLC tiles) |
| ORCA36 | C-grid | 1/36° | $Z^*$ (75 levels) | Global (12962×9173) |
| FIO-COM32 | B-grid | 1/32° | $Z^*$ (57 levels) | Global (11520 × 5504) |
| HYCOM25 | C-grid | 1/25° | Hybrid (41 $p$ & isopycnal) | Global (9000 × 7055) |

**Table B3.** The equation of state (EOS), surface boundary layer (SBL) parametrization used, and tidal forcing in each simulation. *Jackett and McDougall (1995, JMD95) in HYCOM is implemented with the approximation by Brydon et al. (1999). The potential densities were computed following each EOS with the reference pressure of 0 dbar (Fernandes, 2014; Abernathey, 2020; Firing et al., 2021). The EOS for FIO-COM32 is available on Github (https://github.com/roxyboy/swot_adac_ogcms/tree/notebook; a DOI will be allocated upon acceptance of the manuscript). Note that FESOM-GS and ORCA36 do not have tidal forcing whilst the others have at least the leading 5 tidal forcings.

| Simulation | EOS for density | SBL parametrization | Tidal forcing |
|---|---|---|---|
| eNATL60 | TEOS10 | TKE | $M_2$, $S_2$, $N_2$, $O_1$, $K_1$ |
| GIGATL | JMD95 | $\kappa$-$\epsilon$ closure w/ Canuto A formulation | $M_2$, $S_2$, $N_2$, $K_2$, $K_1$, $O_1$, $P_1$, $Q_1$ |
| HYCOM50 | JMD95* | KPP | $M_2$, $S_2$, $N_2$, $K_2$, $K_1$, $O_1$, $P_1$, $Q_1$ |
| FESOM-GS | EOS80 | KPP | N/A |
| LLC4320 | JMD95 | KPP | Full lunisolar tidal forcing |
| ORCA36 | EOS80 | GLS | N/A |
| FIO-COM32 | preTEOS10 | KPP & non-breaking wave induced mixing | $M_2$, $S_2$, $N_2$, $K_2$, $K_1$, $O_1$, $P_1$, $Q_1$ |
| HYCOM25 | JMD95* | KPP | $M_2$, $S_2$, $N_2$, $O_1$, $K_1$ |

**Table B4.** The bathymetry configuration of each simulation.

| Simulation | Bathymetry |
|---|---|
| eNATL60 | Unsmoothed two-min. Etopo2 file of the National Geophysical Data Center. |
| GIGATL | SRTM30plus is smoothed using a Gaussian kernel, w/ a width of 4 grid points. Then another step (to avoid pressure gradient errors) is to check that the steepness of the topography does not exceed $r_{max} = \Delta H / H \leq 0.2$ (cf. Le Corre et al., 2020). |
| HYCOM50 | Nearest $5 \times 5$ box average of the 15-sec. GEBCO_2019 global dataset to each grid point. Then smoothed once w/ a 1-2-1 9 pt. smoother except within 2 grid points of land. |
| FESOM-GS | RTopo-2 (Schaffer et al., 2016). Two smoothing cycles by averaging closest grid points. |
| LLC4320 | Unsmoothed Smith and Sandwell (1997) Version 14.1 & IBCAO Version 2.23. |
| ORCA36 | Two paths of Shapiro filter on Etopo08, upon which it is remapped (w/ bi-linear interpolation) onto model grid. |
| FIO-COM32 | A Blackman radial filter (following Arbic et al., 2004) w/ filter radius of about 7 km is used to smooth the GEBCO dataset. |
| HYCOM25 | Nearest $5 \times 5$ box average of 30-sec. GEBCO_08 20091120 global dataset to each grid point. Then smoothed once w/ a 1-2-1 9 pt. smoother except within 2 grid points of land. |

**Table B5.** The advection and dissipation scheme used for each simulation. Note that some models have biharmonic viscosities and others do not.

| Simulation | Advection scheme (momentum / tracer) | Dissipation scheme (momentum / tracer) |
| --- | --- | --- |
| eNATL60 | $3^{rd}$ order upwind flux form / $3^{rd}$ order upwind TVD | Horizontal laplacian / laplacian iso-neutral |
| GIGATL | $3^{rd}$ order upstream biased flux form / Split and rotated $3^{rd}$-order upstream biased | N/A (achieved implicitly via adv. scheme) |
| HYCOM50 | $2^{nd}$ order FCT flux form / $2^{nd}$ order FCT | Laplacian & biharmonic / laplacian |
| FESOM-GS | $3^{rd}$-$4^{th}$ order FCT flux form / $3^{rd}$-$4^{th}$ order FCT | Biharmonic (flow aware) |
| LLC4320 | Vector invariant form / $7^{th}$ order monotonicity preserving | Biharmonic modified Leith / vertical laplacian |
| ORCA36 | $3^{rd}$ order UBS flux form / 4th order FCT | Horizontal laplacian / laplacian iso-neutral |
| FIO-COM32 | $2^{nd}$ order centered flux form / MDPPM | Biharmonic |
| HYCOM25 | $2^{nd}$ order FCT flux form / $2^{nd}$ order FCT | Laplacian & biharmonic / laplacian |

**Table B6.** The atmospheric and the inclusion of atmospheric pressure variation at the surface.

| Simulation | Atmospheric forcing | Atmos. pressure variation (inverse barometer correction) |
|---|---|---|
| eNATL60 | 3-hourly, ERA-interim (DFS5.2) w/ absolute & relative wind stress | Yes (No) |
| GIGATL | Hourly, CFSR using a bulk formulation w/ relative wind stress | No (No) |
| HYCOM50 | Climatological ERA-40 + 3-hourly wind anomalies from NOGAPS w/ absolute wind stress | No (No) |
| FESOM-GS | JRA55-do-v1.4.0 | No (No) |
| LLC4320 | 6-hourly, 0.14° ECMWF analysis starting in 2011 | Yes (No) |
| ORCA36 | 3-hourly, ECMWF IFS system w/ absolute wind stress, 0.14° | Yes (Yes) |
| FIO-COM32 | 3-hourly, NCEP GFS w/ relative wind stress, 0.25° | Yes (Yes) |
| HYCOM25 | 3-hourly, NAVGEM w/ relative wind stress, 0.5° | Yes (No) |