# Peer review of "Cloud-based framework for inter-comparing submesoscale permitting realistic ocean models"

_Geoscientific Model Development, 2022_

## Referee Comment (RC2)

Review of gmd_2022_27: **Cloud-based framework for inter-comparing submesoscale permitting realistic ocean models** by Takaya Uchida and others

Review by Mike Bell

Summary of paper: This paper outlines a new approach to generating analysis-ready cloud-optimised (ARCO) data whose purpose is to enable open-source scientific analysis of large ocean / atmosphere data sets. It uses this approach to inter-compare @8 high resolution ocean model simulations in the Gulf Stream separation region with a focus on the sea surface height fields and the vertical buoyancy fluxes by sub-mesoscale mixed-layer eddies (SMLE).

This paper is generally well-written and presents some interesting results. It is also part of an important pioneering effort to improve the intercomparison of large data sets. So I expect to recommend that it is accepted for publication after some revision. In intercomparisons of this sort it is important to help the reader to work out what are the most significant scientific results. I make a number of suggestions below that I hope will help the authors to improve their presentation in this regard. I also ask some questions about the ARCO approach which relate to its sustainability. There is finally a list of minor points. In these "grammar" indicates something is not quite right in the sentence; the problem hinges on the word I've picked out.

**Presentation of results (assisting reader to more easily grasp the main scientific points)**

1. L103 and L108-109: Some more comments on Appendix A at this stage would assist the reader. For example
- Say that most models are spun up for only 12-18 months. Would it be possible to show HYCOM50 data after a 12-18 month spin-up? (were the data required to do that archived?) This would reduce the inhomogeneity of the data.
- Could Table A1 include the date of the start of the spin-up and the start and end of the analysis period (or at its least its length)? This is needed for the SSH analysis
- Table A2: were some of the bathymetries smoothed or edited more than others?
- Table A3: note that vertical resolution as well as horizontal resolution varies significantly between the models
- Table A4: is the flux form (rather than vector form) for momentum the default? If the vector form is used is a Hollingsworth correction the default? Please be explicit about this (this is relevant to the vorticity plot)
- Table A4: note that some models have biharmonic viscosities and others do not (relevant to vorticity plot)
- Table A5: note that FESOM-GS and ORCA36 do not have tidal forcing whilst the others have at least the leading 5 tidal forcings.
2. L104: Is it not possible to discuss the differences that are all too visible in Figure 2 in more detail? Couldn't you include an analysis of the time-scale of viscous damping in the models near the grid-scale to see to what extent that could account for the results?
3. Colour scales in figures 2, 3 and 5: Is it possible to use a colour scale which has a somewhat better dynamic range. I can only really make out 5 colours; dark red/blue, light red/blue, white.
4. L128-129: It seems to me that Figure C1 is more scientifically interesting than Figure 3 as the differences due to tidal forcing differences are reduced and there is a comparison with AVISO data. Useful intercomparisons of this sort can be a lot of work. More careful removal of the tides might give a more interesting comparison.

5. L133-134: "interesting": this is a necessary preliminary step in assessing the std deviation. Can you group the models into tidally forced and unforced or indicate "no tides" on the title of the panel for ORCA36 and FESOM-GS?
6. L139: "applying tidal forcing". Another possibility is to remove tides from the SWOT data (as is routinely done for altimeter data). Atmospheric surface pressure forcing is relevant as well as tidal forcing. It's only when the tidal and mesoscale interact that both need to be treated within one model.
7. L140-154: I haven't managed to work out what scientifically valuable points can be extracted from Figure 4. Most of the points made relate to differences in the forcing. There are probably too many lines on the plots. Perhaps you could separate the winter plot into two plots each with 4 lines. Do you compare the winter and summer plots in the text?
8. Section 3.2 seems to be carefully done and a good analysis though I have a question about equation (2) – see minor points.
9. L209-212: The sub-mesoscale buoyancy flux diagnosed from the models is relatively large in FIO-COM32. It's quite hard for the reader to compare the models because the scales on the ordinate in Figure 6 are different in each of the panels.
10. L259: See comment on L139.
11. Figure D1: negative values of C(t) puzzled me. This implies that just occasionally the best fit is obtained using a negative C for the whole domain. I think you calculate C by minimising the square differences of the parametrised and actual fluxes. One would usually plot C(t)*MLI on the x-axis. The slope of the scatter fit would then be shallower than the 1:1 line. It would be informative to give some correlation coefficients. Figure D1 map plots: The scales on some map plots are twice those on others. This makes intercomparison difficult.

**Sustainability of ARCO methodology**

In order to understand the ARCO methodology, I read the Stern et al. 2022 (S22) paper. That paper is well organised and very helpful. The summary of it in section 2 complements it well, being significantly shorter, and comprehensible on its own.

S22 section 4.1 mentions that one of the lessons (re-)learnt from your SWOT exercise was that data transfers between sites is slow and hence geographical proximity is important. Does the S22 approach need to be adjusted in recognition of this point?

I have no expertise in cloud computing. But there are some basic principles of data access that seem to me to be quite generic. The largest data sets have to be stored on cheaper forms of data storage like cartridges that are slow to load up on a system. If a data set is spread across 100s of cartridges access to the data will be slow/expensive. So to be analysis-ready, data has to be sub-setted in a way that suits the type of analysis that will be performed. The data might need to be laid down in perhaps 2 or 3 different ways to enable a wide range of analyses. It's not clear to me what sub-setting approaches you have taken. Is the restriction of the data to the GS region and the time period you have chosen sufficient? For example, for the 3D data required for the SMLE analysis, was the data laid down in a way that was tailored to your analysis – or was it small enough to be stored on disk? It seems to me that the sustainability of the ARCO approach may depend on astute or pragmatic solutions to these issues.

Many readers will have similar questions in their mind when they read the paper. If the previous paragraph contains invalid assumptions it would be useful to point them out so that the proposed approach is better understood. The co-authors include the main authors on S22, and the paper is

not particularly long, so discussion of such points might be within the scope of your paper. The discussion of sustainability otherwise seems lacking in depth.

L236: 100 Euros per month: is this the total cost (that seems unlikely!) or the cost per user given the quoted 64 cores and 256 GB of memory?

**Minor points**

L18: "each party": sometimes one group (often an independent group) is elected to do the analysis

L22: grammar: "needed"

L28: grammar: "by"

L42-43: As I understand it from Stern et al 2022, the data were collected in one place and this was a slow step. In principle they could have been stored close to their origin - but in ARCO formats. I suspect this would create its own difficulties.

L84: "which unifies the API" could you say "which unifies the API to read and load the data" ?

L103: The tables are in appendix A not appendix B.

L112-115: You might consider re-writing this to avoid saying SSH is also known as ADT. The SWOT ADT (not the SSH) needs to be compared with the model's SSH.

L119-120: You can calculate standard deviation of a from $\overline{a^2} - (\bar{a})^2$ so don't really need access to the temporal dimension for this diagnostic.

L129: grammar: tend -> tends

L180: grammar: were

L199-203: I'm not sure I follow this sentence. I believe that the square of the horizontal buoyancy gradient is expected to scale with $(\Delta s)^{-1}$ and there should be a factor $\Delta s / L_f$ in (2) where $L_f$ is a "modified mixed-layer Rossby radius". I think the analysis that has been performed must take this into account.

L269: grammar; "were"

---

## Author Comment (AC1)

We thank the editor Dr. Farneti in handling our manuscript, and Dr. Griffies, Dr. Bell, Dr. Hogg and Dr. Hirschi for their positive and constructive comments. We have acknowledged their work in the Acknowledgements section. Please find our point-by-point reply below in red text.

**Referee #1 (Stephen Griffies)**

This is an enjoyable piece of work that documents a tremendous and exciting advance in our ability to analyze ocean models. I fully support publication and offer only minor comments. We thank the reviewer for his positive comments.

- Line 63: The phrase "we more often than not do not possess" is very awkward. How about "commonly, we do not possess..."
  Adopted.

- Line 112-113: I did not find "absolute dynamic topography" in Gregory et al (2019) paper. Even if ADT is the name used by AVISO, please do connect directly to the now-standard nomenclature in Gregory et al. Furthermore, note that "dynamic topography" is a deprecated term listed in Section 8 of Gregory et al, with three recommended replacements depending on the context. So again, please move to the new nomenclature to avoid confusion.
  While we understand the reviewer's concerns, the terminology of 'Absolute Dynamic Topography' is the one used by AVISO and the naming of their products so we have kept it in the manuscript to be consistent with the AVISO product we are using. (Please also see our reply below). We have strengthened the specific references to ADT and modeled SSH as approximations to the ocean dynamic sea level, to be maximally consistent with Gregory et al. (2019).

- Line 115: where precisely in Gregory et al (2019) are you pointing to? Again, I do not recall us defining "absolute dynamic topography" in Gregory et al, though perhaps I am missing something. And again, "dynamic topography" is not a recommended term since it has multiple meanings depending on the science community.
  We have attempted to clarify the term by adding that 'absolute dynamic topography' is also referred to as ocean dynamic sea level. Specifically, we have changed the paragraph in lines 114-119 as: "In light of the SWOT mission, the primary variable of interest is the ocean dynamic sea level. The AVISO estimate of this quantity is called the Absolute Dynamic Topography (ADT), while the closely related model diagnostic is their Sea Surface Height (SSH) after correcting for the inverse barometer effect if atmospheric pressure variability was simulated. Technically, SSH is defined as the geodetic height of the sea surface above the reference ellipsoid, while ocean dynamic sea level (or ADT) is defined relative to the geoid, but in models where the geoid and reference ellipsoid coincide these two definitions are in practice the same (Gregory et al., 2019). Furthermore, in the specific comparisons made here, a regional average of the ocean dynamic sea level estimates is removed first, so that large-scale, slow changes (e.g., ice sheet contributions) are excluded from the comparison."

- Figure 2: Some model grid spacing is given in km and others in degrees. In the caption, or in Table A3, it would be useful to see a common approach. Additionally, please provide the number of grid points in the domain in Table A3; i.e., the "resolution" as it is normally meant, say, for a computer screen.
  While we understand the reviewer's points, grid spacings in some simulations were indeed defined via km instead of degrees (e.g. GIGATL, FESOM-GS). We have kept the descriptions consistent to the simulations used.
  We have added the number of grid points in Table B2.

- Line 133: "interesting". But I think it is "expected", right? If unexpected, then comment.
  We agree with the reviewer and have added: "... while expected, it is interesting…"

- Figure 4: I failed to find information about the geographical location of this frequency power spectrum.
  We have added in the caption: "The frequency periodograms were computed every ~10 km in Region 1 and then spatially averaged."

**Reference**

- Gregory, J. M., Griffies, S. M., Hughes, C. W., Lowe, J. A., Church, J. A., Fukimori, I., Gomez, N., Kopp, R. E., Landerer, F., Le Cozannet, G. and others. (2019) Concepts and terminology for sea level: Mean, variability and change, both local and global. *Surveys in Geophysics*. doi:10.1007/s10712-019-09525-z;

---

## Author Comment (AC2)

We thank the editor Dr. Farneti in handling our manuscript, and Dr. Griffies, Dr. Bell, Dr. Hogg and Dr. Hirschi for their positive and constructive comments. We have acknowledged their work in the Acknowledgements section. Please find our point-by-point reply below in red text.

**Referee #2 (Mike Bell)**

Summary of paper: This paper outlines a new approach to generating analysis-ready cloud-optimised (ARCO) data whose purpose is to enable open-source scientific analysis of large ocean / atmosphere data sets. It uses this approach to inter-compare @8 high resolution ocean model simulations in the Gulf Stream separation region with a focus on the sea surface height fields and the vertical buoyancy fluxes by sub-mesoscale mixed-layer eddies (SMLE).

This paper is generally well-written and presents some interesting results. It is also part of an important pioneering effort to improve the intercomparison of large data sets. So I expect to recommend that it is accepted for publication after some revision. In intercomparisons of this sort it is important to help the reader to work out what are the most significant scientific results. I make a number of suggestions below that I hope will help the authors to improve their presentation in this regard. I also ask some questions about the ARCO approach which relate to its sustainability. There is finally a list of minor points. In these "grammar" indicates something is not quite right in the sentence; the problem hinges on the word I've picked out. We thank the referee for his thorough review of our manuscript.

**Presentation of results (assisting reader to more easily grasp the main scientific points)**

1. L103 and L108-109: Some more comments on Appendix A at this stage would assist the reader. For example
    a. Say that most models are spun up for only 12-18 months. Would it be possible to show HYCOM50 data after a 12-18 month spin-up? (were the data required to do that archived?) This would reduce the inhomogeneity of the data.
    Regarding HYCOM50, it was spun up from rest and integrated for 20 years. Sensitivity experiments were performed starting from year 15. It took a minimum of 5 years to reach mechanical equilibrium (Chassignet and Xu, 2017).
    For the first 5 years of the HYCOM50 integration, only saved monthly fields were saved. Daily afterwards. We have added this in the table caption.
    We note that Fig. 2 is for illustration purposes and not an in-depth model comparison since the models all have differences in their setup. The integration time of 12-18 months would suggest that the large-scale features from the other simulations are still sensitive to their respective initial conditions.

    b. Could Table A1 include the date of the start of the spin-up and the start and end of the analysis period (or at its least its length)? This is needed for the

SSH analysis

For the sake of storage, only three months for summer (Aug., Sep., Oct.) and winter (Feb., Mar., Apr.) are saved from an arbitrary year per simulation, which are then used for the SSH analysis. We have added the year of outputs used to Table B1.

   c. Table A2: were some of the bathymetries smoothed or edited more than others?
We have added this information to Table B4.

   d. Table A3: note that vertical resolution as well as horizontal resolution varies significantly between the models
Added in the table caption.

   e. Table A4: is the flux form (rather than vector form) for momentum the default? If the vector form is used is a Hollingsworth correction the default? Please be explicit about this (this is relevant to the vorticity plot)
We have added this information to the table.

   f. Table A4: note that some models have biharmonic viscosities and others do not (relevant to vorticity plot)
Added in the table caption.

   g. Table A5: note that FESOM-GS and ORCA36 do not have tidal forcing whilst the others have at least the leading 5 tidal forcings.
Added in the table caption.

2. L104: Is it not possible to discuss the differences that are all too visible in Figure 2 in more detail? Couldn't you include an analysis of the time-scale of viscous damping in the models near the grid-scale to see to what extent that could account for the results?
This is indeed an important point and a likely culprit for the difference we see in Fig. 2. We would like to leave a more in-depth examination for a subsequent paper where we discuss the effect of numerics on the resolved submesoscale flow. Here, we have kept the focus of the manuscript on the implementation and application of the Pangeo Forge framework and showcased a few example diagnostics.

3. Colour scales in figures 2, 3 and 5: Is it possible to use a colour scale which has a somewhat better dynamic range. I can only really make out 5 colours; dark red/blue, light red/blue, white.
We have changed the colormap of the standard deviation in Fig. 3 to purple/orange to differentiate from the blue/red colormap used for demonstrating the mean.

4. L128-129: It seems to me that Figure C1 is more scientifically interesting than Figure 3 as the differences due to tidal forcing differences are reduced and there is a comparison with AVISO data. Useful intercomparisons of this sort can be a lot of work. More careful removal of the tides might give a more interesting comparison.
While we agree that the comparison with AVISO is interesting (i.e. Fig. C1), in light of

SWOT, we would like to keep Fig. 3 in the main text to highlight the importance of including tidal forcing in numerical simulations.

5. L133-134: "interesting": this is a necessary preliminary step in assessing the std deviation. Can you group the models into tidally forced and unforced or indicate "no tides" on the title of the panel for ORCA36 and FESOM-GS?
We have added "no tides" in the titles.

6. L139: "applying tidal forcing". Another possibility is to remove tides from the SWOT data (as is routinely done for altimeter data). Atmospheric surface pressure forcing is relevant as well as tidal forcing. It's only when the tidal and mesoscale interact that both need to be treated within one model.
To our knowledge, the accurate removal of tidal forcing is an area of on-going research (particularly for the non-phase-locked (incoherent) part of the internal tide signals; Zaron and Ray, 2018; Carrere et al., 2021). The benefit of having tidally forced simulations is that we can develop and test such methods of removing tides. We have noted this in line 149.

7. L140-154: I haven't managed to work out what scientifically valuable points can be extracted from Figure 4. Most of the points made relate to differences in the forcing. There are probably too many lines on the plots. Perhaps you could separate the winter plot into two plots each with 4 lines. Do you compare the winter and summer plots in the text?
We have split the spectra plots so as to enlarge them.
We have also added in the text in lines 165-171: "It is interesting to note that at time scales of O(1-10 days), most runs show higher variability during winter than summer (Figure 4a,c), while the tidally forced runs show higher variability at time scales shorter than O(1 day) during summer (Figure 4b,d). The seasonality at time scales shorter than O(1 day) is reversed for ORCA36, a run with no tidal forcing. It is possible that the increase in forward cascade of energy stimulated by the tides are the culprit for higher SSH variability at time scales shorter than the inertial frequency during summer than winter for the tidally forced runs and vice versa for the non-tidally forced runs (Barkan et al., 2021). The overall higher SSH variability at time scales longer than the inertial frequency during winter than summer, on the other hand, is likely due to wind-driven inertial waves (Flexas et al., 2019)."

8. Section 3.2 seems to be carefully done and a good analysis though I have a question about equation (2) – see minor points.
Please see our reply to the referee's minor point below.

9. L209-212: The sub-mesoscale buoyancy flux diagnosed from the models is relatively large in FIO-COM32. It's quite hard for the reader to compare the models because the scales on the ordinate in Figure 6 are different in each of the panels.
We agree that the flux is relatively large in FIO-COM32. Unifying the y-axes would mean that we would have the same axes for all simulations as the axis used for FIO-COM32. This would unfortunately make the temporal fluctuations of HYCOM50 difficult to observe. We have added in the caption: "Note that the y axes vary depending on the magnitude diagnosed from each simulation in order to highlight its

temporal variability."

10. L259: See comment on L139.
    Please see our reply corresponding to L139.

11. Figure D1: negative values of C(t) puzzled me. This implies that just occasionally the best fit is obtained using a negative C for the whole domain. I think you calculate C by minimising the square differences of the parametrised and actual fluxes. One would usually plot C(t)*MLI on the x-axis. The slope of the scatter fit would then be shallower than the 1:1 line. It would be informative to give some correlation coefficients. Figure D1 map plots: The scales on some map plots are twice those on others. This makes intercomparison difficult.
    C(t) is never plotted in Fig. D1 (now Fig. 7) and is always positive (cf. blue lines in Fig. 6). It is calculated by taking the spatial median of C(t,x,y) diagnosed at each grid point by taking the ratio of MLI and the actual flux. This is described in lines 225-226 as: "We diagnosed Ce by taking the ratio between the right-hand and left-hand side of equation (2) at each grid point and time step, and then the horizontal spatial median of it."
    The aim of the map plots is to exhibit the actual buoyancy flux and its equivalent predicted from the MLI parametrization within each model. If we were to unify the scaling, this would make some panels saturate while making it difficult to see the signal for others. We have increased the size of the figure and added in its caption: "Note that the range of colorbar differs depending on the magnitude diagnosed from each model to highlight their spatial features and comparison between the submesoscale buoyancy flux and its equivalent predicted from the parametrization per simulation."

**Sustainability of ARCO methodology**

In order to understand the ARCO methodology, I read the Stern et al. 2022 (S22) paper. That paper is well organised and very helpful. The summary of it in section 2 complements it well, being significantly shorter, and comprehensible on its own.

S22 section 4.1 mentions that one of the lessons (re-)learnt from your SWOT exercise was that data transfers between sites is slow and hence geographical proximity is important. Does the S22 approach need to be adjusted in recognition of this point?
We agree with the referee that in light of limited storage (which is always a constraint in storing and distributing model outputs), some astute planning is needed beforehand to decide what variables and which regions are to be stored.

I have no expertise in cloud computing. But there are some basic principles of data access that seem to me to be quite generic. The largest data sets have to be stored on cheaper forms of data storage like cartridges that are slow to load up on a system. If a data set is spread across 100s of cartridges access to the data will be slow/expensive. So to be analysis-ready, data has to be sub-setted in a way that suits the type of analysis that will be performed. The data might need to be laid down in perhaps 2 or 3 different ways to enable a wide range of analyses. It's not clear to me what sub-setting approaches you have taken. Is

the restriction of the data to the GS region and the time period you have chosen sufficient? For example, for the 3D data required for the SMLE analysis, was the data laid down in a way that was tailored to your analysis – or was it small enough to be stored on disk? It seems to me that the sustainability of the ARCO approach may depend on astute or pragmatic solutions to these issues.

Many readers will have similar questions in their mind when they read the paper. If the previous paragraph contains invalid assumptions it would be useful to point them out so that the proposed approach is better understood. The co-authors include the main authors on S22, and the paper is not particularly long, so discussion of such points might be within the scope of your paper. The discussion of sustainability otherwise seems lacking in depth.

As the referee correctly points out, the amount of data to be stored on the cloud is limited by the funding allocated to the cloud storage. The daily averaging of 3D data (as opposed to hourly outputs) was indeed done due to storage constraints. While we have focused the sub-setting to regional data that correspond to a few SWOT Crossover regions, we believe a 10x10 degree subset over the upper 1000m allows for various analyses to be conducted. For example, the dataset was not tailored for the SMLE analysis where storage of hourly outputs would have allowed for further analysis on how the interaction between inertia- and internal-gravity waves and submesocale flows would have affected the SMLE parametrization. We have noted in the manuscript in line 82: "... (due to cloud storage constraints)."

L236: 1000 Euros per month: is this the total cost (that seems unlikely!) or the cost per user given the quoted 64 cores and 256 GB of memory?
We have added more details on the total cost in lines 266-271 as: "Currently as of writing, the cloud storage provided by OSN is funded by an NSF grant acquired by the Climate Data Science Laboratory at Columbia University, and the JupyterHub on Google Cloud Platform by Centre National d'Études Spatiales (CNES) funding. The cost of cloud resources for the JupyterHub with parallelized computation adds up to roughly 1000 € per month with the maximum computational resources of 64 cores and 256 gigabytes of memory per user; the resources scale on-demand, while the cost of operating the scalable Kubernetes infrastructure is managed by a vendor (2i2c) for a few thousand dollars a month. Although this may seem expensive…"

**Minor points**

- L18: "each party": sometimes one group (often an independent group) is elected to do the analysis
  We have added: "(often an independent group)".

- L22: grammar: "needed"
  Corrected.

- L28: grammar: "by"
  Corrected.

- L42-43: As I understand it from Stern et al 2022, the data were collected in one place and this was a slow step. In principle they could have been stored close to their origin

- but in ARCO formats. I suspect this would create its own difficulties.
We believe that having the ARCO data storage centralized facilitates the managing of data.

- L84: "which unifies the API" could you say "which unifies the API to read and load the data" ?
Adopted.

- L103: The tables are in appendix A not appendix B.
We have corrected the table referencings to Appendix B. Appendix A describes the Pangeo Forge recipes.

- L112-115: You might consider re-writing this to avoid saying SSH is also known as ADT. The SWOT ADT (not the SSH) needs to be compared with the model's SSH.
In response to referee #1's comments, we have rephrased this as: "In light of the SWOT mission, the primary variable of interest is the ocean dynamic sea level. The AVISO estimate of this quantity is called the Absolute Dynamic Topography (ADT), while the closely related model diagnostic is their Sea Surface Height (SSH) after correcting for the inverse barometer effect if atmospheric pressure variability was simulated. Technically, SSH is defined as the geodetic height of the sea surface above the reference ellipsoid, while ocean dynamic sea level (or ADT) is defined relative to the geoid, but in models where the geoid and reference ellipsoid coincide these two definitions are in practice the same (Gregory et al., 2019). Furthermore, in the specific comparisons made here, a regional average of the ocean dynamic sea level estimates is removed first, so that large-scale, slow changes (e.g., ice sheet contributions) are excluded from the comparison."

- L119-120: You can calculate standard deviation of a from $\overline{a^2} - (\overline{a})^2$ so don't really need access to the temporal dimension for this diagnostic.
While we agree with the referee, $\overline{a^2}$ is not always a saved variable as model output.

- L129: grammar: tend -> tends
Corrected.

- L180: grammar: "were"
Corrected.

- L199-203: I'm not sure I follow this sentence. I believe that the square of the horizontal buoyancy gradient is expected to scale with $(\Delta s)^{-1}$ and there should be a factor Δs/Lf in (2) where Lf is a "modified mixed-layer Rossby radius". I think the analysis that has been performed must take this into account.
Our understanding is that the $\Delta s$ factor is there to account for weaker resolved buoyancy gradients when the model resolution is coarse. In other words, the resolved buoyancy gradient at $O(0.1°)$ is different/weaker than the coarse-grained fields of buoyancy resolved at $O(1/50°)$.
Quoting from Fox-Kemper et al. (2011): "*The MLE parameterization (5) is*

*proportional to the horizontal density gradient, a quantity that depends strongly on horizontal resolution. Coarser models have weaker gradients than finer, and sparser observations have weaker gradients than denser. Additionally, the MLE parameterization in (5) is based on one resolved front, rather than a sea of statistically-distributed fronts of varying strength and orientation. Fortunately, one can scale for these effects based on an analysis of the horizontal wavenumber spectrum of near-surface density variance. The Δs/Lf factor in (6) is the result of this analysis (Section 2.1.3). This rescaling can be done with some confidence, as the same near-surface density variance spectrum is found in observations (Section 2.1.1) and in model hierarchies designed to study the effects of differing resolution (Section 2.1.2). "*

As the model outputs we use are all submesoscale permitting, we have left the scaling factor ($\Delta s$) out from our analyses. We have noted this as: "$\Delta s$ was omitted due to all our model outputs partially resolving the submesoscale buoyancy flux. Furthermore, as $\Delta s$ doesn't vary much among the models, this factor would not contribute much to the overall differences between models, in comparison to the greater variability due to numerics, etc., this manuscript is meant to introduce." in lines 222-224.

- L269: grammar; "were"
  Corrected.

**Reference**

- Barkan, R., Srinivasan, K., Yang, L., McWilliams, J.C., Gula, J. & Vic, C. (2021). Oceanic mesoscale eddy depletion catalyzed by internal waves. *Geophysical Research Letters*. doi:10.1029/2021GL094376;
- Carrere, L., Arbic, B.K., Dushaw, B., Egbert, G., Erofeeva, S., Lyard, F., Ray, R.D., Ubelmann, C., Zaron, E. & Zhao, Z., et al. (2021). Accuracy assessment of global internal-tide models using satellite altimetry. *Ocean Science*. doi:10.5194/os-17-147-2021;
- Chassignet, E. & Xiaobiao, X. (2017) Impact of Horizontal Resolution (1/12 to 1/50) on Gulf Stream Separation, Penetration, and Variability. *Journal of Physical Oceanography*. doi:10.1175/jpo-d-17-0031.1;
- Flexas, M.M., Thompson, A.F., Torres, H.S., Klein, P., Farrar, J.T., Zhang, H. & Menemenlis, D. (2019). Global estimates of the energy transfer from the wind to the ocean, with emphasis on near-inertial oscillations. *Journal of Geophysical Research: Oceans*. doi:10.1029/2018JC014453;
- Gregory, J. M., Griffies, S. M., Hughes, C. W., Lowe, J. A., Church, J. A., Fukimori, I., Gomez, N., Kopp, R. E., Landerer, F., Le Cozannet, G. and others. (2019) Concepts and terminology for sea level: Mean, variability and change, both local and global. *Surveys in Geophysics*. doi:10.1007/s10712-019-09525-z;
- Fox-Kemper, B., Danabasoglu, G., Ferrari, R., Griffies, S.M., Hallberg, R.W., Holland, M.M., Maltrud, M.E., Peacock, S. & Samuels, B.L. (2011). Parameterization of mixed layer eddies. III: Implementation and impact in global ocean climate simulations. *Ocean Modelling*. doi:10.1016/j.ocemod.2010.09.002;

- Zaron, E.D. & Ray, R.D. (2018). Aliased tidal variability in mesoscale sea level anomaly maps. *Journal of atmospheric and oceanic technology*. doi:10.1175/JTECH-D-18-0089.1;

---

## Author Comment (AC3)

We thank the editor Dr. Farneti in handling our manuscript, and Dr. Griffies, Dr. Bell, Dr. Hogg and Dr. Hirschi for their positive and constructive comments. We have acknowledged their work in the Acknowledgements section. Please find our point-by-point reply below in red text.

**Referee #3 (Andy Hogg)**

This paper advocates for a cloud-based strategy to address the problems in sharing and analysing the large volumes of data that emerge from high-resolution ocean model simulations. I found the paper to be interesting and well-written, and concur with two previous reviewers that this manuscript is a worthwhile contribution to the literature. I have some minor comments, which the authors may like to take into account, listed below. I would be happy to recommend publication if these issues are addressed. We thank the referee's positive and constructive comments.

- Line 8-9 Consider deleting the sentence naming the 5 models from the abstract? Done.
- Section 2 I found this description of the process of sharing data, and the ARCO format, to be particularly useful. But one thing I don't understand is whether the authors are arguing the Zarr files produced here are optimal for all operations. For example, if I wanted to filter with FFTs, average in time or average in space, would the Zarr chunking remain optimal for all of these operations? Or is there a trade-off between operations? We indeed advocate for the Zarr format for all datasets where parallelized analyses are anticipated. Zarr format scales much better (depending on the computational architecture, up to orders of magnitude) than the NetCDF format for parallelized computation without any trade off.
- Line 115 Improved GS separation is a nice feature, but the global ocean is bigger than just the North Atlantic and there are many more processes revealed by resolution than WBC separation. I'm not convinced that separation is more "key" than other processes that are improved with resolution. Maybe just back away from this statement a little?

We have added: "in the North Atlantic" in the sentence acknowledging that the GS separation is not a global feature.

- Line 122 "this will..." is a little ambiguous.
  We have rephrased this as: "the 3D diagnostics will".
- Line 146 my recollection is that the tides in LLC4320 had a bug in the tidal forcing which overestimates the tidal magnitude (but I apologise that I can't put my hands on the appropriate reference). I suggest the authors check on this issue as they revise the manuscript.

The reviewer is correct. We have added in lines 159: "Also note that tidal forcing in the LLC4320 simulation was inadvertently overestimated by a factor of 1.1121."

- Line 180 "… the two …" also ambiguous.
  We have rephrased this as: "between the submeso- and meso-scale".
- Line 191 there is a case made about daily-averaged submesoscale fields, but it wasn't clear (to this reader) where these daily-averaged fields were used in this paper?
  The 3D diagnostics are based on the daily-averaged fields. We have added this as: "... using the daily-averaged outputs" in line 173.
- Line 222 "This presents ..." ambiguous ... We have rephrased this as: "The smaller predicted values presents..."
- Figure 6 On a first read, I was amazed at the similarities between the parameterised submesoscale fluxes and measured buoyancy flux. Actually, it looked too good to be believable. But when I looked at D1 the comparison was underwhelming. I suspect the use of the spatial median in Fig 6 is unfairly favouring the comparison. I would prefer the authors to show D1 as the main figure, or perhaps show both in the main text, for a warts-and-all comparison of the parameterisation.
  We have put both figures 6 and D1 (now Figure 7) in the main text. We have also added in the Conclusions that the agreement between the submesoscale flux and its prediction from the parametrization are "in the spatially averaged sense" in line 310.
- Section 5 The authors make some good points here and I agree with most of them. But I found the approach to be slightly evangelical. Fundamentally, the argument seems to be "we have found the best approach, but if the scientists/funders don't back us then it will fail". I agree that the approach espoused here is good, and I would like to advocate for it myself. But a more dispassionate discussion of the pros and cons would probably be an advantage here. For example, a significant disadvantage here is the risk that the Google Cloud Platform is discontinued or unavailable to researchers in some nations, for whatever reason. That is not such an outlandish proposition, but could be catastrophic for an open platform like this. There are other risks of equal access, long term funding, etc. I am just asking here for a more objective analysis of the risks here — which would be a greater service to the reader than the advocative approach.

A core design principle of both Pangeo Forge and (IIUC) JupyterHub is being cloud vendor agnostic. So while the JupyterHub for this project happened to be hosted on GCP, and the data happened to be hosted by AWS (i.e. OSN), there is nothing about the underlying technologies — Pangeo Forge and Jupyter — which require these vendors. Unless the future comes down to no major cloud vendor providing such services, we believe the cloud-based framework to be robust.

The JupyterHub that 2i2c runs is also intentionally designed to be cloud-agnostic and none of that technology is dependent on Google. This is codified in 2i2c's "Right to Replicate" principles: https://2i2c.org/right-to-replicate/, and by the constraint that all of the technology behind the hub is community-driven and vendor- and platform-agnostic. We have noted this in lines 279-284 as: "We would like to note that while we have chosen GCP and OSN for the cloud platform, the core design principle and technology behind Pangeo Forge and JupyterHub operated by 2i2c are non-proprietary and cloud vendor agnostic (for example, as defined in 2i2c's "Right to Replicate",

https://2i2c.org/right-to-replicate). We could re-deploy the entire cloud platform on a

different cloud provider with relative ease. This lets the users of this platform benefit from the flexibility and efficiency of the cloud, while minimizing the risk of lock-in and dependence on proprietary technology."

---

## Author Comment (AC4)

We thank the editor Dr. Farneti in handling our manuscript, and Dr. Griffies, Dr. Bell, Dr. Hogg and Dr. Hirschi for their positive and constructive comments. We have acknowledged their work in the Acknowledgements section. Please find our point-by-point reply below in red text.

**Referee #4 (Joel Hirschi)**

I think this manuscript is a most interesting and timely illustration of how storage and analysis of large model datasets may evolve in the coming years. The latest generation of ocean (and also of atmosphere) models now routinely produce datasets of O(Tera-Petabytes). The storage and analysis of these datasets is a major challenge – which often results in cutting edge simulations being underexploited. Here the authors use the cloud-based framework proposed by the Pangeo project to produce an intercomparison of a set of 8 submesoscale-permitting ocean models. The manuscript shows the potential of the cloud framework and provides an assessment of the mixed-layer instability (MLI) parameterisation of Fox-Kemper (2011) across a set of submesoscale-permitting models. The manuscript is clear and well-written and will be an excellent contribution to GMD. I only have a few minor points listed below that might benefit from some clarification.
We thank the reviewer for his supportive comments.

**Comments:**

1. The main motivation of the study is to demonstrate a framework for the intercomparison and analysis of datasets of O(Tera- Petabytes). However, the region of focus around the Gulf Stream separation is actually quite small (~1000 km x 1000 km) and the size of the datasets will be Gigabytes rather than Terabytes or more. The choice to focus on the Gulf Stream separation region is well motivated as this is a region where SWOT tracks will cross. Nevertheless, I wonder if something can be said about how easily the system would scale if comparison and analysis were extended to e.g. the largest domain (North Atlantic) that all 8 models have in common or to global analyses (ie. when the amount of data indeed gets in the order of Petabytes…). Could OSN handle this amount of data? Could it be uploaded onto the cloud within a reasonable amount of time?
In terms of technology, OSN and the Pangeo pipeline are capable of handling petabytes of data. We have uploaded the 4 regions shown in Figure 1 to OSN and the process scaled well. We have added this in lines 88-89 as: "The entire process of zarrifying the data, fluxing them to OSN and cataloging scaled well for the four regions shown in Figure 1." The primary limitation comes from the acquisition of funding to support and maintain such storage on the cloud.

2. I found the MLI assessment most interesting as it adds an interesting piece of science and the agreement seen in Figures 6 and D1 is surprisingly good. However, I am not sure that the explanation given in Appendix D as to why the histogram values shown in Figure D1 are falling under the one-to-one line is correct. Isn't this rather the consequence of taking the spatial median? If the local (i.e. for each grid cell) values are taken for Ce, there is by construction a perfect alignment of the histograms with the one-to-one lines. Any departure from the one to one lines has

therefore to result from summarising the spatial variability with one value (i.e. Ce(t, x,y) --> Ce(t)). I also note here that across the models the slope for the histograms is steeper than the one-to-one lines. As before, I feel that the slope will be affected depending on which value you chose for Ce (e.g. median, mean, mode, 1st, 3rd quartile…etc). Depending on which value you pick and on the distribution of the values Ce(t,x,y), I expect that the histogram values can move above, onto, or below the one-to-one line and that the slope can increase or decrease. What do the distributions of values Ce(t,x,y) actually look like? It might be nice to see an example. This distribution may be a useful guide for deciding on the value of the efficiency coefficient Ce.

If we permit Ce to take spatially negative values, then the reviewer is correct in that "there is by construction a perfect alignment of the histograms with the one-to-one lines" as it is the ratio between the submesoscale buoyancy flux ($w'b'$) and its value predicted from the parametrization. Namely, Ce(t,x,y) is indeed locally negative where frontolysis dominates as $w'b' < 0$ whereas the parametrization by construct can only take positive values (Fig. E1). While Ce is a tuning parameter, energetic consistency of the parametrization requires it to take only positive values (Fox-Kemper et al., 2011). Although the spatial median was chosen so that Ce(t) is less sensitive to spatial extrema, and in order to have the time series of $\overline{w^s b^s}^z$ and the parametrization to agree in the spatially averaged sense, we agree that the histogram would depend on the values chosen for Ce. We have added the histogram for 'Ce's diagnosed as the spatial mean and mode in Appendix E. While the joint histogram seems to align closer to the one-to-one line when the spatial mode is used than the median (Fig. E1), the parametrization tends to overestimate the submesoscale buoyancy flux (Fig. E2). We, therefore, recommend the usage of spatial medians in estimating Ce.

**Details:**

- Figure 5: I suggest to label the panels with w, wm, ws and b, bm, bs.
  Done.

- Figures 6, D1: Use "Ce" rather than "C".
  Done.

**Reference**

- Fox-Kemper, B., Danabasoglu, G., Ferrari, R., Griffies, S.M., Hallberg, R.W., Holland, M.M., Maltrud, M.E., Peacock, S. & Samuels, B.L. (2011). Parameterization of mixed layer eddies. III: Implementation and impact in global ocean climate simulations. *Ocean Modelling*. doi:10.1016/j.ocemod.2010.09.002;

---

## Referee Report (RR1)

Second Review of GMD_2022_27: **Cloud-based framework for inter-comparing submesoscale permitting realistic ocean models** by Takaya Uchida and others

Review by Mike Bell

**Summary**

The authors have replied satisfactorily to most of the technical points in my review. However the replies to my questions on the sustainability of the ARCO methodology were weak and did not respond very directly to my questions (other than the last one) in the normal way. More importantly the manuscript is largely unchanged in this section so the questions that I suggested many readers would have unanswered in their minds when reading the original draft are not answered by the revised version. Andy Hogg also asked that the discussion on the sustainability issue include a more objective discussion of the pros and cons and the reply on that point was similarly weak. As this issue lies at the heart of the paper, the authors really need to go through another round of revisions focused on this part of the paper. Providing the authors give a reasoned and objective response on this issue I would still expect to recommend that the paper is accepted.

**More detailed points**

These more detailed notes refer to the sections and paragraph numbers of the original review.

**Presentation of results**

1. The responses on this point are very thorough and the paper has been revised appropriately.
2. Yes, a detailed examination of this issue could be quite extensive. I'm willing to accept that it is outside the scope of this paper.
3. The authors have changed the colour scale in figure 3 but the dynamic range is no better. So I still think that the colour scales in figures 2, 3 and 5 could be improved but will not insist that they are.
4. The authors give a reason for keeping the figures as they are. This is a minor presentational point so I am content with the response.
5. The paper has been revised appropriately.
6. I'm concerned that the results presented do not properly support the statement "Considering the difference between simulations tidally forced and not, it is likely that in order to emulate the upcoming SWOT observations, applying tidal forcing is a key aspect in addition to model resolution (Savage et al., 2017a, b; Arbic et al., 2018)" and that this could be quoted out of context. It is clear that to compare with SWOT or altimeter data, tidal forcing and atmospheric pressure loading need to be taken into account. But there are pros and cons to doing this interactively. Do the references given supply that evidence? The text that has been added in line 149 does not fit into the rest of the paragraph properly so needs to be revised.
7. Both the figure and supporting text have been improved.
8. Fine
9. The authors make a very valid point in response. But it would be helpful to use a common range ($8 \times 10^{-9}$) for 3 models (GIGATL, HYCOM50 and FESOM-GS ) and $1.2 \times 10^{-8}$ for eNATL60 and LLC4320. The Ce range for GIGATL could be made 0.07 to be in line with eNATL60 and FESOM-GS. I hope these minor changes would be easy to do. They would facilitate comparison.
10. This point was worded slightly differently from the one I questioned in point 6 above. This wording is OK.

11. The additional explanation of the calculation of C(t) is an important addition and the additional plots were requested by other reviewers. I still think that one would usually plot C(t)*MLI on the x-axis. The slope of the scatter fit would then be shallower than the 1:1 line – which is what one expects to see when the fit is not particularly good. If this is relatively easy to do the authors should do that.

**Sustainability of ARCO methodology**

As I said in the summary, the responses for this section didn't respond directly to most of my questions. More importantly only very minor changes to the wording of the paper have been made in response to my comments. The same questions would come up in my mind reading the revised paper as the original one. Could you produce a revised version of this section?

On the point about 1000 Euros per month I still don't really understand what this means. How many users can be supported for 1000 Euros per month and how does the cost scale as the number of users and the number of ARCO data sets are increased?

**Minor points**

These are fine. I just suggest that on line 17 "each party of interest (often an independent group)" is changed to "each of the interested parties (or an independent group)".

---

## Author Response (AR2)

We thank Dr. Bell for his careful reviews. Please see our point-by-point reply below in red text.

**Summary**

The authors have replied satisfactorily to most of the technical points in my review. However the replies to my questions on the sustainability of the ARCO methodology were weak and did not respond very directly to my questions (other than the last one) in the normal way. More importantly the manuscript is largely unchanged in this section so the questions that I suggested many readers would have unanswered in their minds when reading the original draft are not answered by the revised version. Andy Hogg also asked that the discussion on the sustainability issue include a more objective discussion of the pros and cons and the reply on that point was similarly weak. As this issue lies at the heart of the paper, the authors really need to go through another round of revisions focused on this part of the paper. Providing the authors give a reasoned and objective response on this issue I would still expect to recommend that the paper is accepted.

We apologize for not having addressed the referee's previous concerns thoroughly. We hope our replies below and edits to the manuscript have satisfactorily addressed them.

**More detailed points**

These more detailed notes refer to the sections and paragraph numbers of the original review.

**Presentation of results**

1. The responses on this point are very thorough and the paper has been revised appropriately.
   Thank you.

2. Yes, a detailed examination of this issue could be quite extensive. I'm willing to accept that it is outside the scope of this paper.
   Thank you.

3. The authors have changed the colour scale in figure 3 but the dynamic range is no better. So I still think that the colour scales in figures 2, 3 and 5 could be improved but will not insist that they are.
   We have changed the colormap for Figure 2. Regarding Figures 3 and 5, we have tried to choose color schemes which increase monotonically in color saturation and brightness in order to be friendly to readers with color-vision deficiency (cf. Crameri et al., 2020).

4. The authors give a reason for keeping the figures as they are. This is a minor presentational point so I am content with the response.
   Thank you.

5.  The paper has been revised appropriately.
    Thank you.

6.  I'm concerned that the results presented do not properly support the statement "Considering the difference between simulations tidally forced and not, it is likely that in order to emulate the upcoming SWOT observations, applying tidal forcing is a key aspect in addition to model resolution (Savage et al., 2017a, b; Arbic et al., 2018)" and that this could be quoted out of context. It is clear that to compare with SWOT or altimeter data, tidal forcing and atmospheric pressure loading need to be taken into account. But there are pros and cons to doing this interactively. Do the references given supply that evidence? The text that has been added in line 149 does not fit into the rest of the paragraph properly so needs to be revised.
    We have added three more citations which address the importance of tidally-forced simulations in examining the eddy-wave interaction, which SWOT is expected to observe. We have also changed the sentence to: "The difference we find between simulations tidally forced and not is consistent with previous studies which argue that in order to emulate the upcoming SWOT observations, applying tidal forcing is a key aspect in addition to model resolution…"

7.  Both the figure and supporting text have been improved.
    Thank you.

8.  Fine
    Thank you.

9.  The authors make a very valid point in response. But it would be helpful to use a common range (8 10-9) for 3 models (GIGATL, HYCOM50 and FESOM-GS ) and 1.2 10-8 for eNATL60 and LLC4320. The Ce range for GIGATL could be made 0.07 to be in line with eNATL60 and FESOM-GS. I hope these minor changes would be easy to do. They would facilitate comparison.
    Done.

10. This point was worded slightly differently from the one I questioned in point 6 above. This wording is OK.
    Thank you.

11. The additional explanation of the calculation of C(t) is an important addition and the additional plots were requested by other reviewers. I still think that one would usually plot C(t)*MLI on the x-axis. The slope of the scatter fit would then be shallower than the 1:1 line – which is what one expects to see when the fit is not particularly good. If this is relatively easy to do the authors should do that.

    As the actual diagnosed submesoscale flux $(\overline{w^s b^s})$ takes both signs but C(t)*MLI only takes positive values, we argue that having $\overline{w^s b^s}$ on the x axes makes the figure easier to read with the one-to-one line plotted against axes with logarithmic scaling.

**Sustainability of ARCO methodology**

As I said in the summary, the responses for this section didn't respond directly to most of my questions. More importantly only very minor changes to the wording of the paper have been made in response to my comments. The same questions would come up in my mind reading the revised paper as the original one. Could you produce a revised version of this section?

Regarding the referee's previous point: "*The largest data sets have to be stored on cheaper forms of data storage like cartridges that are slow to load up on a system. If a data set is spread across 100s of cartridges access to the data will be slow/expensive. So to be analysis-ready, data has to be sub-setted in a way that suits the type of analysis that will be performed.*", the Zarrification of the model outputs to optimize them for cloud computing can be considered as subsetting the data. The step of Zarrification, however, can be omitted if the outputs were directly saved in Zarr format. As for a case in which no subsetting was applied, we can point to the example of the LLC4320 analysis. Their entire, hourly 3D outputs are stored on the NASA Ames supercomputer and made publicly accessible via their data portal. While the bandwidth of the portal was indeed a bottleneck in analyzing the LLC4320 data, we were still able to apply our cloud-based parallelized analyses to it in a consistent manner with the other model outputs. This exemplifies that terabytes of data can be systematically analyzed without specific subsetting, albeit the efficiency being dependent on the data format (i.e. LLC4320 data is stored in binary format).

As the referee also previously noted, geographical proximity between the data and cloud-computing resources is indeed important.

The management of the ECCO data portal is outside of Pangeo Forge and is done independently by NASA so we have added the points above in Section 2.1 (lines 94-98) as: "Regarding LLC4320, the data were accessed via the ECCO data portal. While there was no particular sub-setting applied to their dataset prior to analyses, the data portal and cloud-based JupyterHub being within geographical proximity (within the U.S.) facilitated the data access. The combination of `llcreader` of the `xmitgcm` Python package to access their data in binary format (as opposed to NetCDF) also enhanced the efficiency (Abernathey, 2019; Abernathey et al., 2021)."

Please also see the "Changes-to-manuscript.pdf" document enclosed in our revision.

On the point about 1000 Euros per month I still don't really understand what this means. How many users can be supported for 1000 Euros per month and how does the cost scale as the number of users and the number of ARCO data sets are increased?

We have added: "... adds up to roughly 1000€ per month for up to three simultaneous full-time users… As of writing, we have consumed 3.5 tera hours of CPU and 92.1 terabytes of RAM monthly on average."

The scaling of cost is dependent on the deal negotiated between the party of interest and Google Cloud Platform (GCP). Here, our contract with GCP allowed for roughly 1000€ per month as of writing. We have added this in line 291 as: "(We note that the operational cost somewhat depends on the contract negotiated amongst the party of interest, GCP and 2i2c.)"

Regarding the cloud storage of ARCO data sets, all of the major cloud providers have public dataset programs to support free hosting of scientific data. For this reason, the cost of storage for this type of data is not exactly a commodified product with discrete unit pricing (like hard drives might be). For Pangeo Forge, the OSN storage allocation is part of the

project grant, which currently is not associated with monetary expense for the storage. We have re-wrote the lines 278-282 as: "Currently as of writing, the JupyterHub on Google Cloud Platform (GCP) is funded by a Centre National d'Études Spatiales (CNES), grant acquired by the MultiscalE Ocean Modeling (MEOM) group at the Institut de Géosciences de l'Environnement, and the operational cost of fluxing data to the OSN cloud storage by an NSF grant acquired by the Climate Data Science Laboratory at Columbia University. (The OSN storage itself allocated to Pangeo Forge is not associated with monetary expense nor any egress fees; https://www.openstoragenetwork.org/get-involved/get-an-allocation/.)"

**Minor points**

These are fine. I just suggest that on line 17 "each party of interest (often an independent group)" is changed to "each of the interested parties (or an independent group)".
Adopted.

**Reference**

- Abernathey, R.P. Petabytes of Ocean Data, Part I: NASA ECCO Data Portal. https://medium.com/pangeo/petabytes-of-ocean-data-part-1-nasa-ecco-data-portal-81e3c5e077be, 2019.
- Abernathey, R.P. et al. `xmitgcm`: Read MITgcm mds binary files into xarray. doi:10.5281/zenodo.596253, 2021.
- Crameri, F., Shephard, G.E. & Heron, P.J. The misuse of colour in science communication. Nature Comm. doi:10.1038/s41467-020-19160-7, 2020.